# Optimal and Efficient Link Insertion for Hitting-Time Minimization

## Abstract

We study the computational problem of strategically adding links to a graph to minimize the hitting time between two group of nodes. Our problem has applications for machine learning tasks and for social network analysis, including graph neural network design and bridging polarized groups with opposite views in a network. Formally, we are given a graph where the set of nodes is partitioned into two disjoint groups, $R$ and $B$, and we assume a *random-walk* process modeling navigation over the graph. Our goal is to add a given number of edges to the graph to minimize the expected number of steps to encounter a node in $B$ starting from nodes in $R$, via the random walk. While the problem is generally **NP**-hard, we show that when the random walk starts from the stationary distribution over the induced subgraph of $R$, the problem becomes *optimally solvable in polynomial-time*, and we present an extremely efficient *optimal* greedy strategy. Remarkably, our method applies to both directed and undirected graphs, and many widely-adopted random-walk models, for example, PageRank.

Our experimental evaluation demonstrates that our method outperforms state-of-the-art baselines for similar metrics. Remarkably, our method achieves up to four orders of magnitude of speedup compared to existing methods, scaling to networks with millions of edges, which cannot be processed with current methods.

## 1 Introduction

What are the best links to add to a graph to reduce the separation between two groups represented by two disjoint set of nodes? This question can be of interested in many settings such as option discovery for Markov decision processes (Jinnai et al., 2019; 2020), graph neural network design (Arnaiz-Rodriguez et al., 2022; Black et al., 2023), mitigating polarization in social networks by recommending cross-group connections (Musco et al., 2018), improving navigability in hyperlink networks by suggesting bridging links (Haddadan et al., 2021; Menghini et al., 2019), or minimizing exposure to harmful content by facilitating access to diverse perspectives (Coupette et al., 2023; Fabbri et al., 2022). To formally model our question, we rely on the notion of hitting time—the expected number of steps for a random walk starting from one group of nodes to reach the other—as a principled measure of group separation. Our goal is to identify the *optimal* link insertions to minimize hitting times.

The problem of adding the best $b$ links in a graph to minimize the hitting time between two groups of nodes has been recently studied in the literature (Adriaens et al., 2023; Haddadan et al., 2021; Coupette et al., 2023; Fabbri et al., 2022). Since the general problem is **NP**-hard (Adriaens et al., 2023; Haddadan et al., 2021; Coupette et al., 2023; Fabbri et al., 2022), existing approaches are approximate, yielding often impractical algorithms and suboptimal solutions. Furthermore, while real-world graphs are mostly directed, current methods offer theoretical guarantees only for undirected graphs (Adriaens et al., 2023).

In this work we make a significant advance to the above line of research. We first observe that minimizing the *average hitting-time*, a typical problem studied in the literature, does not account for the *importance* of different nodes in a group. That is, the average does not capture the *relative importance* of a node in its group, e.g., influential social network users or popular Wikipedia pages.

We thus introduce a variant of the *average hitting-time minimization problem* (HTMP) by considering a weighted-average scheme, in which each node is weighted proportionally to its relative impor-

tance within its own group. The node importance is quantified using the *stationary distribution*. We refer to our new problem variant as the *stationary hitting-time minimization problem* (S-HTMP).

Surprisingly, we can show that, in contrast to HTMP, the S-HTMP problem can be solved *optimally* in *polynomial time*. We develop PARITY, an *optimal* algorithm for the S-HTMP problem, which runs in almost linear time on undirected graphs, and in $\mathcal{O}(n_R^3)$ time on directed graphs, where $n_R$ denotes the size of the group of interest. The latter complexity can be reduced to $\mathcal{O}(n_R^2)$ when an additive approximation is allowed by leveraging well-known approximation techniques; see Lemma 5. Note that prior to our work, no existing method can directly solve the HTMP on directed graphs, making our method PARITY the first method to deal with directed graphs.

In summary, in this paper we make the following contributions.

**1.** We introduce a new variant of the problem of computing $b$ link insertions for a graph to minimize the average hitting time between two disjoint groups of nodes. The main novelty lies in using a weighted average to capture node importance. That is, nodes are weighted proportional to the stationary distribution of a random walk. Our framework can be used with *any* random-walk process (modeling graph explorations, e.g., option discovery for Markov decision processes) having a stationary distribution.

**2.** We develop PARITY, an *optimal* polynomial-time algorithm to solve our novel problem variant for both directed and undirected graphs. We use the stationary distribution to initialize random walks, and achieve our optimality guarantees. To the best of our knowledge, no prior methods are known to optimize directly hitting-time objectives for directed graphs. Surprisingly, we also show that there exit classes of graphs for which our algorithm PARITY outputs an optimal solution to the average hitting-time minimization problem.

**3.** We conduct an extensive empirical evaluation of PARITY. Our approach significantly outperforms previous methods in terms of runtime, *achieving up to four orders of magnitude of speedup*, while achieving competitive or superior results in minimizing different metrics, such as the average or the maximum hitting-time, over groups of interest.

Our code is available for review and reproducibility and will be made public upon acceptance.

## 2 PRELIMINARIES

For $n \in \mathbb{N}$, let $[n] = \{1, \ldots, n\}$. Let $G = (V, E)$ be a graph, with $V = [n]$, and $E \subseteq V \times V$. Let $m = |E|$ be the number of links or *edges* of $G$. Each edge $e = (u, v) \in E$ represents a *directed* edge from node $u$ to $v$. The graph is *undirected* if $(u, v) \in E$ implies $(v, u) \in E$. Let $w = \langle u_1 \ldots u_k \rangle$ be a *walk* from $u_1$ to $u_k$, i.e., a sequence such that $(u_{i-1}, u_i) \in E$, for any $2 \leq i \leq k$.

For a directed graph $G$, we denote with $\delta_G^+(u) = \{v \in V : (u, v) \in E\}$ the *out*-neighborhood of $u \in V$, and with $\delta_G^-(u) = \{v \in V : (v, u) \in E\}$ the *in*-neighborhood. For an undirected graph we define $\delta_G(u) = \delta_G^-(u) = \delta_G^+(u)$. In this paper we assume that the set of nodes $V$ can be partitioned into two disjoint sets, which for convenience we refer to as the set of *red* nodes $R$ and the set of *blue* nodes $B$. In other words, $V = R \cup B$. Without loss of generality, we let $R = [n_R]$. Given $A \subseteq V$, we let $G_A \subseteq G$ be the *induced subgraph* by $A$ such that $G_A = (A, E \cap (A \times A))$, i.e., the subgraph containing only nodes and edges of $A$ from $G$.

We quantify the degree of separation between red and blue nodes using *random walks*. A random walk is a stochastic process $(X_t)_{t \in \mathbb{N}}$ that describes a walk through the vertices of the graph. In a graph with high degree of separation, a random walk starting at a node in $R$ is likely to stay within $G_R$ for a long time before encountering a node in $B$. We will focus on random walks that can be defined as a *Markov chain*, and thus can be associated with a *transition matrix* $\boldsymbol{P} \in \mathbb{R}^{n \times n}$ that captures the conditional transition probabilities of the process $(X_t)_{t \geq 0}$, i.e., $\boldsymbol{P}_{ij} = \Pr[X_{t+1} = j \mid X_t = i]$ for $i, j \in [n]$. For example, in a *simple random walk*, at each node $i$ the next node $j$ is chosen uniformly at random among the out-neighbors of $u$. In our work, we will consider transition matrices $\boldsymbol{P}$ that *depend on the topology of* $G$, focusing on the simple random-walk model, and the (personalized) PageRank (see Sec. 3). A random walk associated with transitions $\boldsymbol{P}$ admits a *stationary distribution* if it exists $\boldsymbol{\pi} \in [0, 1]^n$ such that $\boldsymbol{\pi}^T = \boldsymbol{\pi}^T \boldsymbol{P}$. Our ideas can be easily

adapted to cases where the random walk on graph $G$ is modeled with more nuanced PageRank variants (Bianchini et al., 2005) or weighted random walks (Haddadan et al., 2021).

Our goal is to reduce the separation from nodes in $R$ to nodes in $B$. Similarly to prior work, we quantify this separation using the *hitting time*, i.e., the expected number of steps for a random walk starting at a node in $R$ to reach $B$ for the first time.

**Hitting time.** For $i \in R$, let $T_i(\boldsymbol{P})$ be the expected number of steps for a random walk $(X_t)_{t \geq 0}$ with transition matrix $\boldsymbol{P}$ and starting at node $X_0 = i$ to reach a node in $B$ for the first time. That is, $T_i(\boldsymbol{P}) = \mathbb{E}[\min\{j : X_j \in B\}]$, with $\min\{\emptyset\} = \infty$. We simply write $T_i$ when $\boldsymbol{P}$ is clear from the context. The value $T_i$ is referred to as the *hitting time* for the set $B$ starting from node $i \in R$.

When $\boldsymbol{P}$ is a transition matrix of a random walk over $G$, we denote by $\boldsymbol{P}_R$ the $n_R \times n_R$ top-left block of $\boldsymbol{P}$. For any $i \in R$ and $k \geq 0$, let $Y_{k,i} \doteq \mathbb{1}[\{X_0, \ldots, X_k\} \cap B \neq \emptyset \mid X_0 = i]$. Let $Z_i$ be a random variable denoting the first index $j \geq 0$ such that $X_j \in B$, conditioned on $X_0 = i$. For any $k \geq 0$, we have $Z_i > k$ if and only if $Y_{k,i} = 0$. The hitting time can be expressed as

$$T_i = \sum_{k=0}^{\infty} k \Pr[Z_i = k] = \sum_{k=1}^{\infty} k \left(\Pr[Z_i > k-1] - \Pr[Z_i > k]\right) = \sum_{k=0}^{\infty} \Pr[Z_i > k] . \quad (1)$$

Recall that $Y_{k,i} = 0$ implies that the first $k$ steps of the random walk, starting from $X_0 = i$, with $i \in R$, remain entirely within the nodes in $R$. By considering a modified random walk over $G$ where each node in $B$ becomes an absorbing node (Kemeny et al., 1969), it follows that[1]

$$\Pr[Y_{k,i} = 0] = \boldsymbol{e}_i^T (\boldsymbol{P}_R)^k \boldsymbol{1} . \quad (2)$$

Notably, the definition of $T_i$ does not depend on which specific blue node is first encountered during the random walk, but it only depends on the *time* at which any blue node is reached. This observation, also noted in previous work (Adriaens et al., 2023), is formalized as follows.

**Observation 1.** *Consider $G = (V, E)$ and $e \in (V \times V) \setminus E$. Let $G' = (V, E \cup \{e\})$ be a new graph where $e$ is added to the graph $G$, and let $T_i(e)$ be the hitting time to encounter a node of $B$ in $G'$ starting from $i \in R$. For any fixed $j \in R$, the sequence $(T_i(e))_{i \in R}$ is the same for any $e \in \{(j, \ell) : \ell \in B\} \setminus E$. In other words, the hitting time $T_i(e)$ depends only on the* red *endpoint of $e$ and not on its* blue *endpoint.*

Using Observation 1, link insertions to a graph $G$ of the form $(i, b)$ for $i \in R$ and $b \in B$ can be represented with a non-negative vector of integers $\boldsymbol{x} \in (\mathbb{Z}_{\geq 0})^{n_R}$, where the $i$-th coordinate, $\boldsymbol{x}_i$, denotes the number of links added to the graph $G$ that include node $i \in R$ and arbitrary blue nodes. We refer to $\boldsymbol{x}$ as the *link-insertion* vector. The link-insertion vector $\boldsymbol{x}$ must satisfy the *structural properties* of $G$. For example, if a node $i \in R$ is already connected to *all* blue nodes, no additional links can be added to $i \in R$, and thus $\boldsymbol{x}_i = 0$. Formally, let $\boldsymbol{c} \in (\mathbb{Z}_{\geq 0})^{n_R}$, where the $i$-th entry $\boldsymbol{c}_i$ expresses an upper bound on how many edges can be added to the graph from a node $i \in R$ to a node $b \in B$. We set $\boldsymbol{c}_i = |B| - |E \cap (\{i\} \times B)|$, for each $i \in R$. Finally, for any $i \in R$ and link-insertion vector $\boldsymbol{x}$, we denote with $T_i(\boldsymbol{P}(\boldsymbol{x}))$ the hitting time on the random walk with transition matrix $\boldsymbol{P}(\boldsymbol{x})$ obtained from the graph $G'$ *after* adding edges to $G$ according to $\boldsymbol{x}$.

## 3 PROBLEM FORMULATION

We now introduce the optimization problem that we address in this paper.

Let $b \geq 1$ be an integer corresponding an edge budget, i.e., how many links can be added to graph $G$ in total. Let $\boldsymbol{\alpha} \in \Delta^{n_R - 1}$ be a weighting of the $n_R$ nodes in $R$ in the $(n_R - 1)$-dimensional simplex, and let $\boldsymbol{P}$ be a *transition matrix* of a random walk over $G$.

**Problem 1.** *We define the $(\boldsymbol{P}, \boldsymbol{\alpha}, b)$-hitting-time minimization problem (HTMP) as follows:*

$$\min_{\boldsymbol{x} \in \mathbb{Z}^n} f_{\boldsymbol{\alpha}}(\boldsymbol{x}) \doteq \min_{\boldsymbol{x}} \sum_{i \in R} \boldsymbol{\alpha}_i T_i(\boldsymbol{P}(\boldsymbol{x})) \quad such \ that \quad \boldsymbol{0} \leq \boldsymbol{x} \leq \boldsymbol{c} \quad and \quad \|\boldsymbol{x}\|_1 \leq b . \quad (*)$$

---

[1] We will use $\boldsymbol{e}_i$ to denote vectors of the standard basis of the Euclidean space $\mathbb{R}^{n_R}$, and $\boldsymbol{1} \in \mathbb{R}^{n_R}$ a vector with all its entries equal to 1.

If $\boldsymbol{P}$ models a simple random walk over $G$, and the weighting $\boldsymbol{\alpha}$ is *uniform*, i.e., $\boldsymbol{\alpha} = \boldsymbol{1}/n_R$, the problem reduces to adding $b$ edges to minimize the *average* hitting time of the red nodes—a setting previously studied by Adriaens et al. (2023), who also stated the **NP**-hardness of the problem.

In this work, we require mild conditions for $\boldsymbol{P}$ that enable us to tackle the HTMP problem with a specific instantiation of the vector $\boldsymbol{\alpha}$. Our choice of $\boldsymbol{\alpha}$ provides the following key benefits: (*i*) Equation (∗) becomes *solvable in polynomial time* with a simple greedy approach for *any* transition matrix $\boldsymbol{P}$; (*ii*) the vector $\boldsymbol{\alpha}$ captures the importance of each node $i \in R$ when random walks are restricted to the subgraph $G_R$ induced by the nodes in $R$; and (*iii*) the hitting time is considered with respect to random walk defined by the transition matrix $\boldsymbol{P}$. Specifically, we will study transitions according to simple random walks and personalized PageRank, while as we mentioned before our model can be easily coupled with any other random walk.

Before formally discussing our optimization problem, we present two necessary conditions for the instantiation of our framework to be technically well-defined.

**Condition A.** *(i) The random walk conditioned to $G_R$, converges to a* stationary distribution.[2] *(ii) There exists at least one edge from a red node to a blue node, i.e., $(R \times B) \cap E \neq \emptyset$.*

First note that the above condition (*i*) naturally holds for a PageRank walk model on $G_R$, or its variants extensively studied in the literature (Yang et al., 2024). In contrast, condition (*i*) may not hold for a simple random walk on an arbitrary graph $G_R$. In such a case we have to restrict the HTMP to the *largest strongly connected component* of $G_R$ as done in prior work (Adriaens et al., 2023).[3] Finally, condition (*ii*) is a simple technical requirement, which can be enforced by adding a *single* edge between a node $R$ and a node in $B$, in the unlikely case that none exists.

Consider a random walk $X_0, X_1, \ldots$ restricted uniquely to $G_R$, i.e., yielding a stochastic transition matrix with its rows summing to 1. By Condition A (*i*), the random walk over $G_R$ is associated with a unique stationary distribution $\boldsymbol{\pi}_R \in \Delta^{n_R-1}$ (Mitzenmacher & Upfal, 2017). We define a special case of the $(\boldsymbol{P}, \boldsymbol{\alpha}, b)$-HTMP problem as follows.

**Problem 2.** *We define the stationary hitting-time minimization problem (S-HTMP) with budget $b$ as the special case of the $(\boldsymbol{P}, \boldsymbol{\alpha}, b)$-HTMP problem with input $(\boldsymbol{P}, \boldsymbol{\pi}_R, b)$, that is, $\boldsymbol{\alpha} = \boldsymbol{\pi}_R$, and the random walk on $G_R$ starts from the stationary distribution $\boldsymbol{\pi}_R$ of $G_R$.*

In other words, the S-HTMP requires the addition of at most $b$ links of the form $(i, j)$ for $i \in R$ and $j \in B$ to minimize the expected hitting time of a random walk initialized with the *stationary distribution* $\boldsymbol{\pi}_R$ of the random walk over $G_R$. In this setting, the weight of each node $i$ in $R$ in the objective function may not be uniform anymore, but instead follows the distribution $\boldsymbol{\pi}_R$. This choice is motivated as follows. First, real-world graphs are known to have small *mixing time*.[4] Thus, a random walk over $G_R$, will typically converge to $\boldsymbol{\pi}_R$ after a very small number of steps, regardless of its initial state. Second, the weighting induced by $\boldsymbol{\pi}_R$ can be interpreted as an importance score for $i \in R$ reflecting how often each node $i \in R$ is visited by a random walk whose transitions follow the topology of $G_R$ and are proportional to $\boldsymbol{P}_R$. For example, if $G$ is a web-graph and the walk over $G_R$ follows PageRank-like transitions (see Sec. 4.1), then $\boldsymbol{\pi}_R$ assigns more weight to nodes with high PageRank value. Consequently, pages $i \in R$ with larger $\boldsymbol{\pi}_{R,i}$ are visited more frequently, e.g., such pages are ranked higher by search-engines. Our objective therefore prioritizes reducing the hitting time $T_i$ proportionally to expected number of visits to each node $i \in R$. For these reasons, we believe that the choice of the initialization in Problem 2, provides a more interpretable and practical setting compared to a uniform weighting (as studied by Adriaens et al. (2023)).

In contrast with the general problem HTMP stated to be **NP**-hard, our new problem variant named S-HTMP is computationally tractable.

**Theorem 1.** *There exists a polynomial-time algorithm that solves the stationary hitting-time minimization problem (S-HTMP) with budget $b$.*

---

[2]More precisely, the Markov chain associated to the random walk model yielding $\boldsymbol{P}$, when *conditioned* to the states of $R$ with transitions based on $G_R$ must be irreducible and aperiodic (Mitzenmacher & Upfal, 2017).

[3]A simple alternative approach is to make $G_R$ strongly connected, reducing the budget $b$ for the HTMP.

[4]The number of steps required by a Markov Chain to approximate its stationary distribution.

## 4 METHODS

In this section, we prove Theorem 1 by presenting a polynomial-time greedy algorithm.

### 4.1 AN ANALYTICAL EXPRESSION FOR THE HITTING-TIME OBJECTIVE

Consider the $(\boldsymbol{P}, \boldsymbol{\alpha}, b)$-HTMP problem. Let $\boldsymbol{A}_R$ be the $n_R \times n_R$ adjacency matrix associated to the induced subgraph $G_R \subseteq G$, where for $i, j \in R$, it is $\boldsymbol{A}_{ij} = 1$ if $(i, j) \in E$, and $\boldsymbol{A}_{ij} = 0$ otherwise. We define the diagonal matrix $\boldsymbol{D}(\boldsymbol{x}) \doteq \operatorname{diag}((|\delta_G^+(1)| + \boldsymbol{x}_1)^{-1}, \ldots, (|\delta_G^+(n_R)| + \boldsymbol{x}_{n_R})^{-1})$.[5]

We now express the transition matrix $\boldsymbol{P}_R(\boldsymbol{x})$ of the graph $G'$ obtained after adding the edges as captured by the link-insertion vector $\boldsymbol{x}$ as a function of $\boldsymbol{A}_R$ and $\boldsymbol{D}(\boldsymbol{x})$ for $(i)$ the simple random walk, and $(ii)$ the (personalized) PageRank walk.

$(i)$ For a *simple random-walk* (SW) it is $\boldsymbol{P}_{ij}^{\mathrm{SW}} = \frac{\mathbb{1}[j \in \delta_G^+(i)]}{|\delta_G^+(i)|}$, $i, j \in [n]$, hence $\boldsymbol{P}_R^{\mathrm{SW}}(\boldsymbol{x}) \doteq \boldsymbol{D}(\boldsymbol{x})\boldsymbol{A}_R$.

$(ii)$ Similarly, when considering the (personalized) PageRank walk (PR) we have

$$\boldsymbol{P}_R^{\mathrm{PR}}(\boldsymbol{x}) \doteq \gamma \boldsymbol{P}_R^{\mathrm{SW}}(\boldsymbol{x}) + (1 - \gamma)\mathbf{1}\boldsymbol{a}^T , \tag{3}$$

where $\boldsymbol{P}_R^{\mathrm{SW}}(\boldsymbol{x})$ is a simple random walk with the exception that all rows $i = 1, \ldots, n_R$ of $\boldsymbol{D}(\boldsymbol{x})\boldsymbol{A}_R$ with $\boldsymbol{y}_i^T = \boldsymbol{0}^T$ (i.e., dangling nodes) are replaced with $\frac{1}{n_R + \boldsymbol{x}_i}\mathbf{1}^T$. Further, $\gamma \in (0, 1)$ corresponds to a *damping factor* and $\boldsymbol{a}^T$ corresponds to the *personalization vector*; see Appendix B.

**Lemma 1.** *Let $\boldsymbol{P}$ be a transition matrix satisfying Condition A. Then, the hitting times of the walk on the graph $G'$ augmented with link insertions given by vector $\boldsymbol{x}$, can be written as $T_i(\boldsymbol{P}(\boldsymbol{x})) = \boldsymbol{e}_i^T \left[ \sum_{k \geq 0}(\boldsymbol{P}_R(\boldsymbol{x}))^k \right] \mathbf{1} = \boldsymbol{e}_i^T(\boldsymbol{I} - \boldsymbol{P}_R(\boldsymbol{x}))^{-1}\mathbf{1}$. Furthermore, the objective of the $(\boldsymbol{P}, \boldsymbol{\alpha}, b)$-HTMP problem can be expressed as $f_{\boldsymbol{\alpha}}(\boldsymbol{x}) = \boldsymbol{\alpha}^T(\boldsymbol{I} - \boldsymbol{P}_R(\boldsymbol{x}))^{-1}\mathbf{1}$.*

We can show that $f_{\boldsymbol{\alpha}}(\boldsymbol{x})$ is convex with respect to $\boldsymbol{x} \in \mathbb{R}^{n_R}$. Unfortunately, we cannot rely on the convexity of $f$ to identify an *integer* solution $\boldsymbol{x}$ that minimizes $f$, given the complex dependence of $f_{\boldsymbol{\alpha}}(\boldsymbol{x})$ on $\boldsymbol{x}$. Surprisingly, in the case $\boldsymbol{\alpha} = \boldsymbol{\pi}_R$, the objective function $f_{\boldsymbol{\pi}_R}(\boldsymbol{x})$ simplifies as follows.

**Lemma 2.** *Let $\boldsymbol{\pi}_R$ be the stationary distribution of a random walk over $G_R$ satisfying Condition A. The objective of the* stationary *hitting-time minimization problem for a simple random walk and a (personalized) PageRank walk, respectively, satisfy*

$$f_{\boldsymbol{\pi}_R}^{SW}(\boldsymbol{x}) = (1 - \boldsymbol{\pi}_R^T \boldsymbol{D}(\boldsymbol{x})\boldsymbol{A}_R\mathbf{1})^{-1} \ \ and \ \ f_{\boldsymbol{\pi}_R}^{PR}(\boldsymbol{x}) = \gamma^{-1}(1 - \boldsymbol{\pi}_R^T \boldsymbol{D}(\boldsymbol{x})\boldsymbol{A}_R\mathbf{1})^{-1} . \tag{4}$$

The simplified expressions in Equation (4) provide a significant reduction in complexity compared to the general form of the objective in Lemma 1. Lemma 2 offers an *extremely powerful result*. When $\boldsymbol{\alpha} = \boldsymbol{\pi}_R$, the vector-matrix products can be brought inside the matrix inversion operation.

The intuition behind this result is as follows. Suppose $X_0 \sim \boldsymbol{\alpha}$ for an arbitrary weighting vector $\boldsymbol{\alpha}$. Then, the distributions of the states $\{X_1, \ldots, X_k\}$ conditioned on $\{X_1, \ldots, X_k\} \subseteq R$ may change depending on $G_R$, the subgraph induced by the red nodes. However, if $X_0 \sim \boldsymbol{\pi}_R$, then conditioning on $\{X_1, \ldots, X_k\} \subseteq R$ ensures that the distribution of each state follows $\boldsymbol{\pi}_R$, according to the definition of the stationary distribution. That is, the probability of reaching a blue node for the first time remains *invariant over time* when the random walk follows $\boldsymbol{\pi}_R$.

Since $p(\boldsymbol{x}) \doteq \boldsymbol{\pi}_R^T \boldsymbol{D}(\boldsymbol{x})\boldsymbol{A}_R\mathbf{1}$ is a scalar and the function $(1 - p(\boldsymbol{x}))^{-1}$ is monotone for $p(\boldsymbol{x}) \in (0, 1)$, the stationary hitting-time minimization problem with budget $b$ for both $\boldsymbol{P}_R^{\mathrm{SW}}$ and $\boldsymbol{P}_R^{\mathrm{PR}}$ can be equivalently formulated as:

$$\min_{\boldsymbol{x} \in \mathbb{Z}^n} p(\boldsymbol{x}) \doteq \boldsymbol{\pi}_R^T \boldsymbol{D}(\boldsymbol{x})\boldsymbol{A}_R\mathbf{1} \ \ \text{such that} \ \ 0 \leq \boldsymbol{x} \leq \boldsymbol{c} \ \ \text{and} \ \ \|\boldsymbol{x}\|_1 \leq b . \tag{\ss}$$

### 4.2 PARITY: AN OPTIMAL POLYNOMIAL-TIME ALGORITHM

In this section, we present a polynomial-time algorithm for the S-HTMP problem with budget $b$ (Problem 2) by leveraging our formulation in Equation (\ss). Let $\boldsymbol{M} \doteq \boldsymbol{\pi}_R\mathbf{1}^T\boldsymbol{A}_R^T$ Using the matrix

---

[5]For $\boldsymbol{y} \in \mathbb{R}^z$, $\operatorname{diag}(\boldsymbol{y})$ returns $\boldsymbol{D} \in \mathbb{R}^{z \times z}$ such that $\boldsymbol{D}_{ii} = \boldsymbol{y}_i$ and $\boldsymbol{D}_{ij} = 0$ for $i \neq j, i, j \in [z]$.

$M$ that encodes both the importance given by the vector $\boldsymbol{\pi}_R$ and the structure of the graph $G_R$, the objective $p(\boldsymbol{x})$ can be equivalently written as,[6]

$$p(\boldsymbol{x}) = \boldsymbol{\pi}_R^T \boldsymbol{D}(\boldsymbol{x}) \boldsymbol{A}_R \mathbf{1} = \langle \boldsymbol{M}, \boldsymbol{D}(\boldsymbol{x}) \rangle_F = \sum_{i=1}^{n_R} \frac{\boldsymbol{M}_{ii}}{|\delta_G^+(i)| + \boldsymbol{x}_i} \ .$$

Note that each edge addition over $\boldsymbol{x}_i$ does not affect the value of the other terms for $j \in R, j \neq i$. Leveraging this observation, we design an *optimal* greedy algorithm to minimize $p(\boldsymbol{x})$. The pseudocode for our algorithm, named PARITY, is given in Appendix C, working as follows:

Initially, we set $\boldsymbol{x}_i = 0$ for every red node $i \in R$. We then iteratively allocate each of the $b$ edges, one at a time. At each iteration, for each $i \in R$, we compute the marginal decrease in the objective:

$$\Delta_i = \frac{\boldsymbol{M}_{ii}}{|\delta_G^+(i)| + \boldsymbol{x}_i} - \frac{\boldsymbol{M}_{ii}}{|\delta_G^+(i)| + \boldsymbol{x}_i + 1} \ . \tag{5}$$

We pick the node $i$ with the largest decrease $\Delta_i$ in $p(\boldsymbol{x})$, and update $\boldsymbol{x}_i$ to $\boldsymbol{x}_i + 1$. This step is repeated until either all $b$ edges have been allocated or it is not possible to add any more edges. Inspecting Equation (5), one can observe that each gain $\Delta_i$ is monotonically decreasing with respect to $\boldsymbol{x}_i \geq 0$: there are diminishing returns from adding more edges to the same node. Hence, allocating the next edge to the node currently giving the largest reduction in cost (i.e., $i \in R$ such that $i = \arg\max_j \Delta_j$) is optimal at each step. For efficiency, we also keep track of the different values $\Delta_i$ using a priority queue. The following lemma establishes that this greedy strategy indeed returns an optimal solution.

**Lemma 3.** *The greedy algorithm* PARITY *returns an optimal solution* $\boldsymbol{x}^*$ *to the S-HTMP problem with budget $b$, as described in Equation (‡).*

Next, we also draw a connection between the solution $\boldsymbol{x}^*$ provided by PARITY, and the evaluation of $\boldsymbol{x}^*$ for the HTMP with uniform weights. Let $f_{\text{avg}}(\boldsymbol{x})$ be the objective of the HTMP under uniform weights (i.e., $\boldsymbol{\alpha} = \mathbf{1}/n_R$) for a link-insertion vector $\boldsymbol{x}$. Furthermore, let $d_{TV}(\boldsymbol{z}_1, \boldsymbol{z}_2)$ denote the *total variation distance* between two discrete distributions $\boldsymbol{z}_1, \boldsymbol{z}_2$ defined on a set $\mathcal{Z}$, i.e., $d_{TV}(\boldsymbol{z}_1, \boldsymbol{z}_2) = \frac{1}{2} \sum_{i \in \mathcal{Z}} |\boldsymbol{z}_1(i) - \boldsymbol{z}_2(i)|$. Then the following holds.

**Lemma 4.** *Let $\boldsymbol{x}$ be any solution to the S-HTMP, then $|f_{\boldsymbol{\pi}}(\boldsymbol{x}) - f_{\text{avg}}(\boldsymbol{x})| \leq 2 \max_{r \in R}\{T_r(\boldsymbol{P}(\boldsymbol{x}))\} d_{TV}(\boldsymbol{\pi}_R, \boldsymbol{u}_R)$ where $\boldsymbol{u}_R$ is the uniform distribution over $R$, i.e., $\mathbf{1}/n_R$, and $d_{TV}(\cdot)$ corresponds to the total variation distance. In particular, for undirected graphs it holds that $|f_{\boldsymbol{\pi}}(\boldsymbol{x}) - f_{\text{avg}}(\boldsymbol{x})| \leq \frac{\max_{r \in R}\{T_r(\boldsymbol{P}(\boldsymbol{x}))\}}{2|E_R|} \sum_{r \in R} \left||\delta_{G_R}(r)| - \bar{\delta}_{G_R}\right|$ where $\bar{\delta}_{G_R}$ is the average degree over $G_R$ and $|E_R|$ is number of edges in $G_R$.*

For $\boldsymbol{x} = \boldsymbol{x}^*$ the above lemma quantifies the deviation between S-HTMP and HTMP with uniform weights for an optimal solution $\boldsymbol{x}^*$ provided by PARITY. Our lemma shows that the performance of $\boldsymbol{x}^*$ on the HTMP with uniform weights objective depends on: $(i)$ the total variation distance between $\boldsymbol{\pi}_R$ and the uniform distribution $\mathbf{1}/n_R$; and $(ii)$ $\max_{r \in R}\{T_r(\boldsymbol{P}(\boldsymbol{x}^*))\}$. Term (ii) can be bounded by $2n_R^{3/4} f_{\text{avg}}(\boldsymbol{x}^*)$ using a result by Adriaens et al. (2023, Theorem 2). Let $\boldsymbol{x}_{\text{avg}}^*$ be an optimal solution for $f_{\text{avg}}$, we leave as an open direction identifying tight bounds for term (ii) as a function of $f_{\text{avg}}(\boldsymbol{x}_{\text{avg}}^*)$ (required to bound the approximation ratio $f_{\text{avg}}(\boldsymbol{x}^*)/f_{\text{avg}}(\boldsymbol{x}_{\text{avg}}^*)$). Surprisingly, we observe that the total-variation term (i) shows that there exist classes of graphs for which PARITY achieves an optimal solution to the HTMP under uniform weights, for example $d$-regular undirected graphs $G_R$. We assess empirically the tightness of the upper bound from Lemma 4 in Appendix D.6.

**Time complexity.** PARITY involves three steps: (1) Computing the stationary distribution $\boldsymbol{\pi}_R$ of $G_R$; (2) Computing the diagonal of the matrix $\boldsymbol{\pi}_R \mathbf{1}^T \boldsymbol{A}_R^T$; (3) Greedily allocate the $b$ edges. Step (3) uses a priority queue and requires $\mathcal{O}(b \log n_R)$ time. For Step (2), note that $M_{ii} = \boldsymbol{\pi}_{R,i} \sum_{j=1}^{n_R} A_{ji}$, where $\sum_{j=1}^{n_R} A_{ji}$ is the in-degree of node $i$ in the subgraph $G_R = (R, E_R)$. Hence, computing these diagonal entries requires $\mathcal{O}(n_R + |E_R|)$ time. The more demanding task is the computation of the stationary distribution $\boldsymbol{\pi}_R$. A direct method is to compute the solution using the definition $\boldsymbol{x}^T \boldsymbol{P} = \boldsymbol{x}^T$, which requires up to $\mathcal{O}(n_R^3)$ operations. Alternatively, one can apply the *power method* (Wilkinson, 1988) or more advanced techniques (Cohen et al., 2016). We leverage

---

[6] $\langle \boldsymbol{A}, \boldsymbol{B} \rangle_F$ denotes the Frobenius inner product, i.e., $\text{Tr}(\bar{\boldsymbol{A}}^T \boldsymbol{B})$.

the power method in Sec. 5, which after $q = \Omega\big(\log(1/\varepsilon)/(1 - \lambda_2)\big)$ iterations, yields an estimate $\widehat{\boldsymbol{\pi}}_R$ satisfying $\|\widehat{\boldsymbol{\pi}}_R - \boldsymbol{\pi}_R\|_2 \leq \varepsilon$, where $\lambda_2$ is the second largest eigenvalue of $\boldsymbol{P}_R$. Each iteration costs $\mathcal{O}(\min\{n_R^2, |E_R|\})$, corresponding to the cost of a matrix-vector multiplication.

Finally, if an approximate distribution $\widehat{\boldsymbol{\pi}}_R$ is used in place of the exact $\boldsymbol{\pi}_R$, the optimality guarantee of Algorithm 1 no longer holds. However, as shown in the next lemma, the resulting solution can still be bounded by an *additive approximation*.

**Lemma 5.** *Assume that* PARITY *uses an approximation* $\hat{\boldsymbol{\pi}}_R$ *of the stationary distribution* $\boldsymbol{\pi}_R$ *such that* $\|\hat{\boldsymbol{\pi}}_R - \boldsymbol{\pi}_R\|_2 \leq \varepsilon$, *for some* $\varepsilon > 0$. *Let* $\hat{\boldsymbol{x}}$ *be the output of* PARITY *with* $\hat{\boldsymbol{\pi}}_R$, *and let* $\boldsymbol{x}^*$ *be the optimal solution of Equation* (‡). *Then,* $p(\hat{\boldsymbol{x}}) \leq p(\boldsymbol{x}^*) + 2\varepsilon\sqrt{n_R}$.

Summarizing, we have the following cases:

**1.** For a *directed* graph, the complexity of PARITY is proportional to $\mathcal{O}(n_R^3 + m + \log n_R)$ if the stationary distribution $\boldsymbol{\pi}_R$ of the red nodes is computed exactly. Instead, using the power method to obtain an approximation $\hat{\boldsymbol{\pi}}_R$, PARITY achieves a solution with additive error $\varepsilon$ with respect to an optimal solution for both $\boldsymbol{P}^{\text{SW}}$ and $\boldsymbol{P}^{\text{PR}}$ in time

$$\mathcal{O}\big(\min\{n_R^2, |E_R|\} \log(n_R/\varepsilon)/(1 - \lambda_2) + m + b \log n_R\big) \ .$$

**2.** For an *undirected* graph and $\boldsymbol{P}^{\text{SW}}$, the value $\boldsymbol{\pi}_{R,r}$ of the stationary distribution of a random walk in the red subgraph for each node $r \in R$ is proportional to $\delta_{G_R}(r)$ (Mitzenmacher & Upfal, 2017, Th. 7.13). Thus, it is possible to compute the stationary distribution in $\mathcal{O}(n_R + m)$. In this case, the complexity of Algorithm 1 reduces to $\mathcal{O}(n_R + m + b \log n_R)$. On the other hand, the complexity for $\boldsymbol{P}^{\text{PR}}$ is the same as for directed graphs, given that we have to apply the power method.

To conclude, Theorem 1 follows from Lemma 3 and the above discussion on the time complexity.

# 5 EXPERIMENTAL EVALUATION

We design our experimental evaluation to study the following research questions:

**Q1**. Assess the objective value of the S-HTMP for PARITY compared to the state-of-the-art link-insertion algorithms (Sec. 5.1).
**Q2**. Evaluate the solution of PARITY for established hitting-time metrics, as considered in previous work (Adriaens et al., 2023; Haddadan et al., 2021) (Sec. 5.1).
**Q3**. Assess the efficiency and scalability of PARITY—especially on directed graphs, where no current method can directly minimize the HTMP objective, i.e., Problem 1 (Sec. 5.2).
**Q4**. Assess PARITY on the S-HTMP under PageRank walks with different personalization (deferred to Appendix D.6 for space constraints).

**Baselines and setup.** We compare against the state-of-the-art algorithms for polarization reduction through link insertions, based on hitting-time objectives. We consider two major baselines: (1) `RepBubLik` (Haddadan et al., 2021), which aims to minimize the *bubble radius*, i.e., a function related to the hitting times of nodes in $R$; (2) `Greedy+` (Adriaens et al., 2023), state-of-the-art method for reducing the average and maximum hitting time over nodes in $R$ for a simple random walk. We also consider several other baselines, described in Appendix D.2. Due to the high computational cost of running the baselines (Sec. 5.2) we run each baseline once, similarly we also run PARITY once, given its extremely stable runtime (i.e., less than 1% of variability across different runs). For all baselines, we use their default parameters as specified in the codebase of previous work (Adriaens et al., 2023), see Appendix D.4. A key advantage of PARITY is that, unlike the baselines, it does not require any hyper-parameters. Furthermore, our algorithm is *deterministic*.

To compare against the baselines, we consider *simple random walks* over $G$, i.e., we set $\boldsymbol{P} = \boldsymbol{P}^{\text{SW}}$. Hence, to enforce Condition A, which is also required by Adriaens et al. (2023), we obtain the largest connected component $C_R \subseteq G_R$ of the subgraph $G_R$ induced by the nodes in $R$ (see Table 1 for the size $\kappa_{\max}$ of such component).[7] The largest connected component is then used both to evaluate the objective S-HTMP and as input to PARITY. In Appendix D.6 we consider the entire (directed) graph $G$ as input to PARITY, by considering a PageRank walk over $G_R$.

---

[7] In the directed case we considered the largest *strongly* connected component.

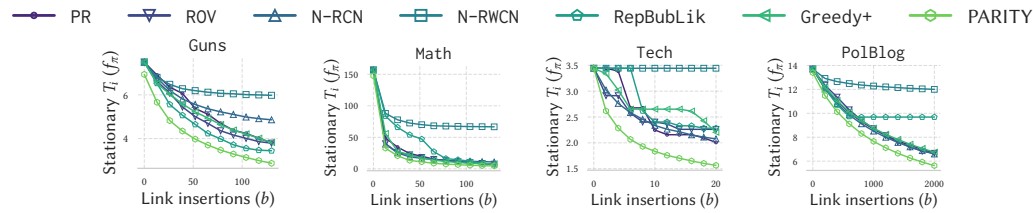

Figure 1: Results for the S-HTMP with budget $b$.

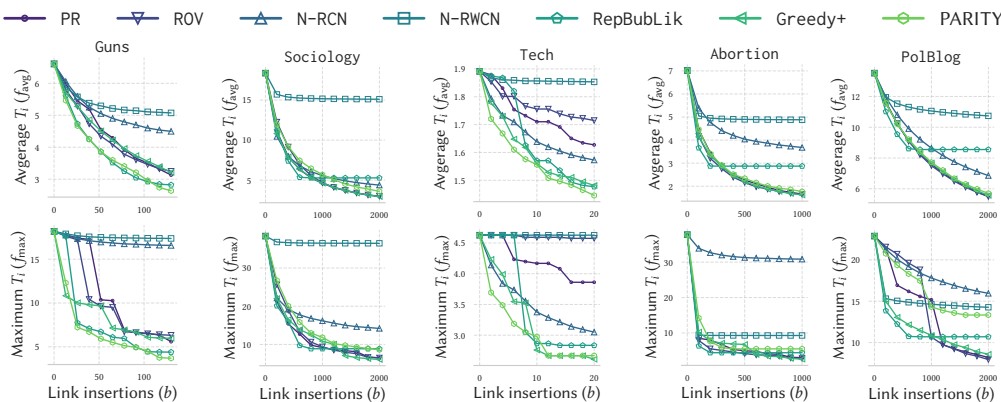

Figure 2: Hitting time values after $b$ link insertions. For each dataset: (top) $f_{\text{avg}}$: the average hitting time over the nodes in $R$; and (bottom) $f_{\text{max}}$: the maximum hitting time over nodes $R$.

**Evaluation metrics.** All methods return a set of links to be added to a graph $G$. To assess the quality of the link insertions we consider the following metrics: the objective function S-HTMP $f_{\boldsymbol{\pi}}$; $f_{\text{avg}} \doteq f_{\boldsymbol{\alpha}}$ for $\boldsymbol{\alpha} = \frac{1}{|R|}\mathbf{1}$, i.e., the *average* hitting time; and $f_{\text{max}} \doteq \max_{i \in R} T_i(\boldsymbol{P}(\boldsymbol{x}))$, i.e., the *maximum* hitting time over $R$. See Appendix D.5 for more details. Here, we fix $\boldsymbol{P} = \boldsymbol{P}^{\text{SW}}$, i.e., we consider simple random walks. The objectives are computed via matrix inversion.

Details on the datasets and additional results are provided in Appendices D.1 and D.6.

## 5.1 HITTING-TIME MINIMIZATION

**Stationary hitting time.** First, we address **Q1** by evaluating the objective of S-HTMP, which we denote by $f_{\boldsymbol{\pi}}$. The results are reported in Fig. 1. As demonstrated in our theoretical analysis, PARITY provides an optimal solution for this problem and thus achieves the smallest value of $f_{\boldsymbol{\pi}}$ compared to baselines. We remark that our baselines are not specifically designed to optimize the S-HTMP. The baselines optimize different objectives that may yield solutions far from the optimal solution for $f_{\boldsymbol{\pi}}$ (see Sec. 6 for more details). As expected, by increasing the budget $b$ the hitting time from the nodes in $R$ to nodes in $B$ always reduces, and in practice a few edge additions (e.g., around 20 to 50) are often sufficient to halve the value of $f_{\boldsymbol{\pi}}$ with respect to the original graph. In the next section we investigate how the link insertions (identified by PARITY and optimal for $f_{\boldsymbol{\pi}}$), behave on previously studied hitting-time metrics.

**Average and maximum hitting time.** Next we answer **Q2** by evaluating edge additions according to the *average* and *maximum* hitting time metrics. The results are reported in Figure 2.

We start by commenting on the results for $f_{\text{avg}}$. For the average hitting-time objective, the solution obtained by PARITY achieves remarkable values, being often the best (or the second best after RepBubLik). That is, the solution of PARITY yields in practice very small values for $f_{\text{avg}}$, on most datasets and values of $b$ considered. Our results show that the stationary distribution over $G_R$ is a very good proxy to identify nodes on which to add new edges, especially for fast-mixing graphs, e.g., real-world ones. Similarly, for $f_{\text{max}}$, we observe that, perhaps very surprisingly, the solution

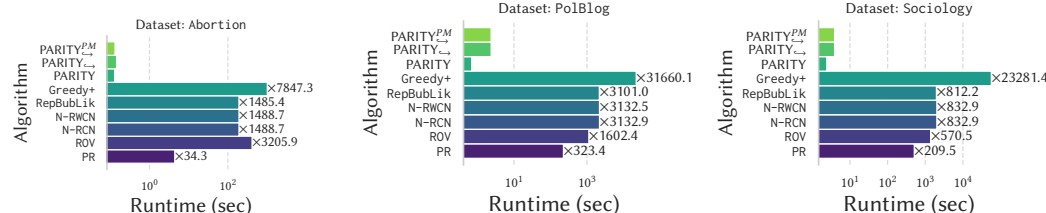

Figure 3: Runtime of all algorithms to process the highest value of $b$ in the setting of Figure 2. We use PARITY$_\hookrightarrow$ to denote the runtime of PARITY if executed on the respective *directed* graph and coupled with an exact computation of the stationary distribution, instead when using the power-method we mark it as PARITY$_\hookrightarrow^{PM}$. We also report the speedup of PARITY over each baseline algorithm on undirected graphs.

of PARITY is among the best ones and it improves over Greedy+, which is specifically designed to optimize this objective. Interestingly, Greedy+ provides a much better solution than PARITY on PolBlog. There are various possible reasons, such as the fact that PolBlog is slowly mixing. This highlights that optimizing $f_{\max}$ could be much more challenging compared with $f_{\text{avg}}$. To further supplement our results, we provide in Appendix D.6, a comparison between the solutions obtained by PARITY, Greedy+ and RepBubLik, showing that PARITY selects significantly distinct nodes $i \in R$ for new link insertions, compared to baselines. That is, the solution in output to PARITY inserts new links to nodes $i \in R$ with higher importance over $G_R$ as captured by $\boldsymbol{\pi}$.

**Summary for Q1 and Q2.** We showed that PARITY yields optimal values for $f_{\boldsymbol{\pi}}$ and can be used as a good proxy to optimize both $f_{\max}$ and $f_{\text{avg}}$. Note that PARITY is the only algorithm that is able to complete its execution for the Politics dataset (see Appendix D.6); we further discuss the performance of the methods in the next section.

## 5.2 EFFICIENCY AND SCALABILITY

In this section we address **Q3** assessing the efficiency and scalability of PARITY both on undirected graphs (compared to the various baselines) and directed graphs.

**Undirected graphs.** To compare all algorithms we consider the setting of Sec. 5.1, measuring the runtime to process the larger values of $b$ used for each dataset. The results are displayed in Figure 3. We observe that our algorithm PARITY consistently outperforms all baselines, achieving: (1) from 2 to 4 orders of magnitude of speedup (corresponding to the ratio between the runtime of the baseline algorithm and the runtime of PARITY) compared to the state-of-the-art method Greedy+, and (2) similar speedups compared to both RepBubLik and ROV. Our algorithm PARITY requires less than ten seconds to complete its execution and optimize $f_{\boldsymbol{\pi}}$, while baseline algorithms may take hours to complete. This should not surprise the reader, as for undirected graphs PARITY runs in almost linear time in $|R|$, while most other baselines optimize complex objective values requiring large computational resources. Supporting the utility of our approach, as demonstrated by the fact that PARITY is the only method able to terminate on the Politics dataset in less than three minutes over a time-limited execution set to two hours.

**Directed graphs.** We show the runtime of PARITY for directed graphs in Figure 3, where we denote with PARITY$_\hookrightarrow$ when PARITY computes the exact stationary distribution and with PARITY$_\hookrightarrow^{PM}$ when we use PARITY and the power-method for 30 iterations to approximate the stationary distribution. We observe that PARITY is very efficient, i.e., several orders of magnitude compared with the baselines over the undirected case. We also note that the running times of the two PARITY versions on directed graphs are very close. This shows that PARITY can be used to minimize hitting times over directed graphs, a problem not practically solvable by previous methods.

**Summary for Q3.** Our results show that PARITY is highly scalable, by completing its execution on large datasets and achieving up to 4 orders of magnitude of speedup compared to existing state-of-the-art methods that optimize similar but more complex objective values. In addition, PARITY provides a simple and effective method to deal with directed graph for which no current practical

methods are available. This makes PARITY a powerful addition to the existing methods for hitting-time minimization, as it can be used to optimize $f_{\boldsymbol{\pi}}$, $f_{\text{avg}}$, and $f_{\max}$ efficiently and with theoretical guarantees as demonstrated in Sec. 5.1. We further test PARITY's scalability on directed and undirected large data never tested before in Appendix D.6.

## 6 RELATED WORK

There is extensive literature on the problem of modifying the structure of a graph to optimize a given graph property, such as graph connectivity, centrality, density, diameter, and more. Here we focus on algorithmic approaches that alter the graph structure to optimize hitting-time-based objectives.

**Link-insertion methods.** Link insertion has been studied as a strategy to mitigate adverse phenomena in graphs, such as polarization and echo chambers, by enhancing connectivity between disparate groups (Colin & Maniu, 2024; Cinus et al., 2023; Chitra & Musco, 2020). Haddadan et al. (2021) define a node-level score based on hitting-times, and the average score yields a global measure of the bias in a graph. They propose greedy algorithms to reduce this bias by adding or swapping $k$ edges, leveraging the monotonicity and submodularity of the objective and sampling approaches for efficiency. Adriaens et al. (2023) study the problem of adding edges to *directly minimize* the hitting-time for a partition of the nodes in two groups, in contrast to earlier works that designed methods to *maximize the reduction* in hitting-time-based objectives. Adriaens et al. (2023) leverage the supermodularity property and existing ideas to optimize supermodular functions. Their methods scale poorly in practice and cannot be used on directed graphs as we show in our experimental evaluation.

**Edge-rewiring methods.** A different approach is to *rewire* a small number of edges to optimize desirable properties (Chan & Akoglu, 2016; Sydney et al., 2013). An edge-rewiring operation swaps the end-points of two existing disjoint edges. Such an operation is suitable specific recommender-systems applications, e.g., "what to watch next" recommendations in video-sharing platforms (Fabbri et al., 2022). Existing works apply greedy strategies for maximizing hitting-time reduction (Fabbri et al., 2022), or for reducing the time spent on a given group of nodes associated with harmful content (Coupette et al., 2023). These works strongly differ from our setting, as we do not consider edge rewirings and we aim to minimize *directly* the hitting-time.

**Polarization metrics**. A rich literature proposes polarization metrics (see the survey by Interian et al. (2023)); notably, Garimella et al. (2018) introduce the random walk controversy (RWC) score, which captures cross-group exposure. Adding links to minimize RWC is a computationally challenging task, our baseline (ROV) for this task is shown to perform poorly in our setting.

**Machine learning.** For machine-learning applications, the optimization of several hitting-time metrics through $b$ edge additions has been studied in the context of Markov Decision Processes (Jinnai et al., 2020; 2019), and GNN over-squashing and over-smoothing prevention (Arnaiz-Rodriguez et al., 2022; Black et al., 2023).

## 7 CONCLUSION

We study the problem of adding links to a graph to minimize the hitting time between two groups, offering a significant contribution that can be used, for example, to mitigate polarization in real-world graphs. Specifically, we introduce a novel problem where nodes are weighted based on the stationary distribution of a random-walk model over a subgraph of interest. Such a weighting scheme captures the importance of the nodes within their group and, in contrast to uniform weighting, enables an efficient polynomial-time optimal solution. Our empirical evaluation shows that our method PARITY compared to state-of-the-art methods: ($i$) achieves a significant runtime improvement (up to $30\,000\times$ of speedup); and ($ii$) outperforms or matches hitting time reduction across different metrics. In addition, our method scales on large graphs and can optimize efficiently directed graphs, for which no previous practical methods were known.

Interesting future directions for our work include: ($i$) obtain non-trivial bounds on the approximation ratio of a solution obtained by PARITY for the average and maximum objectives of the HTMP (i.e., $f_{\text{avg}}$ and $f_{\max}$); ($ii$) identify classes of graphs for which PARITY can provably yield close to optimal solutions to the average HTMP; ($iii$) evaluate the solution computed by our algorithm PARITY for machine learning tasks such as the one mentioned in Sec. 6.

## ETHICS STATEMENT

The approach proposed in this paper aims to reduce separation between disparate groups in graphs by strategically adding links according to our novel problem the S-HTMP. Our goal is to provide a simple and practical tool to bridge groups with different views, reducing polarization and enhancing diversity on the Web and on social media platforms.

Our algorithm, like similar ones found in the literature, could potentially be misused to generate edge recommendations that steer users toward polarizing or radicalized content, rather than its intended purpose of making diverse and healthy recommendations. However, the risk of intentional misuse is no greater than that already present in recommendation algorithms themselves. Additionally, unintentional misuse can be mitigated thorough impact assessment and audits. Therefore, we are confident that, overall, our methods can positively contribute to the well-being of digital platforms.

## REPRODUCIBILITY STATEMENT

The missing proofs of all our theoretical results are available in Appendix A.

We discuss different instantiations for the personalization vector of PageRank walks (see Sec. 4.1) in Appendix B.

The detailed pseudocode of our algorithm PARITY is available in Appendix C.

Our code is made available in the following anonymized repository: `https://anonymous.4open.science/r/optimalHTM`, containing all necessary scripts to reproduce the results of our experimental procedure. In addition, to the scripts available in the source code repository, we provide the following material to further complement our experimental procedure:

- in Appendix D.1 we provide necessary statistics on all datasets used in our experiments. In our source code we provide all the links to download each dataset – given that we used publicly available data.

- in Appendix D.2 we describe additional baseline algorithms (not reported in the main body for space constraints).

- in Appendix D.3 we report the details on the enviroment used to perform all experiments.

- in Appendix D.4 we discuss how we set the parameters of all the algorithms compared, including our algorithm PARITY.

- in Appendix D.5 we provide details on how we assess the efficacy of existing methods for link additions on our new objective value of the S-HTMP.

- in Appendix D.6 we provide additional results for the settings of Secs. 5.1 and 5.2, due to space constraints. In addition, we also provide a discussion for **Q4** (see Sec. 5).

## LLM USAGE

AI tools were used for editing and polishing purposes. Specifically, LLMs were employed for light editing such as grammar checking, typo correction, and minor revisions of author-written text. In addition, we used LLMs to improve the quality of the figures displayed in the experimental section. We did not use LLMs for coding our method, rather than light debugging. All LLM outputs were reviewed, edited, and verified by the authors, who take full responsibility for the final content.

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

APPENDIX

## A    MISSING PROOFS

*Proof of Lemma 1.* Using Equations (1) and (2), we obtain

$$T_i(\boldsymbol{P}(\boldsymbol{x})) = \boldsymbol{e}_i^T \left[ \sum_{k \geq 0} (\boldsymbol{P}_R(\boldsymbol{x}))^k \right] \boldsymbol{1} = \boldsymbol{e}_i^T (\boldsymbol{I} - \boldsymbol{P}_R(\boldsymbol{x}))^{-1} \boldsymbol{1} \,, \tag{6}$$

where the last step is due to the convergence of the Neumann series $\sum_{k \geq 0} (\boldsymbol{P}_R(\boldsymbol{x}))^k$ since $\|\boldsymbol{P}_R(\boldsymbol{x})\|_2 < 1$ from Condition A.[8] Finally using the definition of S-HTMP (i.e., Equation (∗)) we have that

$$f_{\boldsymbol{\alpha}}(\boldsymbol{x}) = \sum_{i \in R} \boldsymbol{\alpha}_i T_i(\boldsymbol{P}(\boldsymbol{x})) = \sum_{i \in R} \boldsymbol{\alpha}_i \boldsymbol{e}_i^T (\boldsymbol{I} - \boldsymbol{P}_R(\boldsymbol{x}))^{-1} \boldsymbol{1} = \boldsymbol{\alpha}^T (\boldsymbol{I} - \boldsymbol{P}_R(\boldsymbol{x}))^{-1} \boldsymbol{1} \,. \tag{7}$$

$\square$

*Proof of Lemma 2.* **Simple random walk.** We first prove our claim for $f_{\boldsymbol{\alpha}} = f_{\boldsymbol{\pi}_R}^{\mathrm{SW}}$. Let $X_0 \sim \boldsymbol{\pi}_R$, and consider the sequence $X_0, X_1, \ldots$ in the graph $G'$ obtained after adding edges to $G$ according to $\boldsymbol{x}$. By definition of $\boldsymbol{\pi}_R$, it must hold that:

$$\Pr\left[X_i = r | \{X_1, \ldots X_{i-1}, X_i\} \subseteq R\right] = \boldsymbol{\pi}_{R,r} \,.$$

By induction, we show that $\Pr\left[\{X_0, X_1, \ldots, X_i\} \subseteq R\right] = \left(\boldsymbol{\pi}_R^T \boldsymbol{D}(\boldsymbol{x}) \boldsymbol{A}_R \boldsymbol{1}\right)^i$ for any $i \geq 0$. The statement is clearly true for $i = 0$. For the induction step, we have

$$\Pr\left[\{X_1, \ldots, X_i\} \subseteq R\right] = \Pr\left[(X_i \in R) \cup (\{X_1, , \ldots, X_{i-1}\} \subseteq R)\right]$$
$$= \Pr\left[X_i \in R \mid \{X_1, \ldots, X_{i-1}\} \subseteq R\right] \cdot \Pr\left[\{X_1, \ldots, X_{i-1}\} \subseteq R\right]$$
$$= \Pr\left[X_i \in R \mid \{X_1, \ldots, X_{i-1}\} \subseteq R\right] \left(\boldsymbol{\pi}_R^T \boldsymbol{D}(\boldsymbol{x}) \boldsymbol{A}_R \boldsymbol{1}\right)^{i-1} \,.$$

We can conclude the argument by evaluating

$$\Pr\left[X_i \in R \mid \{X_1, \ldots, X_{i-1}\} \subseteq R\right] = \sum_{r \in R} \Pr\left[X_i \in R, X_{i-1} = r \mid \{X_1, \ldots, X_{i-1}\} \subseteq R\right]$$
$$= \sum_{r \in R} \Pr\left[X_i \in R \mid X_{i-1} = r\right] \cdot \Pr\left[X_{i-1} = r \mid \{X_1, \ldots, X_{i-1}\} \subseteq R\right]$$
$$= \sum_{r \in R} \boldsymbol{e}_r^T \boldsymbol{D}(\boldsymbol{x}) \boldsymbol{A}_R \boldsymbol{1} \boldsymbol{\pi}_{R,r} = \boldsymbol{\pi}_R^T \boldsymbol{D}(\boldsymbol{x}) \boldsymbol{A}_R \boldsymbol{1} \,.$$

Therefore, we have that:

$$f_{\boldsymbol{\pi}_R^{\mathrm{SW}}} = \sum_{r \in R} \boldsymbol{\pi}_{R,r} T_r(\boldsymbol{P}_R^{\mathrm{SW}}(\boldsymbol{x})) = \sum_{k \geq 0} (\boldsymbol{\pi}_R^T \boldsymbol{D}(\boldsymbol{x}) \boldsymbol{A}_R \boldsymbol{1})^k = (1 - \boldsymbol{\pi}_R^T \boldsymbol{D}(\boldsymbol{x}) \boldsymbol{A}_R \boldsymbol{1})^{-1} \,,$$

proving the first part of the claim.

**Personalized PageRank walk.** We now consider $f_{\boldsymbol{\alpha}} = f_{\boldsymbol{\pi}_R}^{\mathrm{PR}}$. Recall that the transition matrix of the personalized PageRank walk (see Equation (3)) is given by $\boldsymbol{P}_R^{\mathrm{PR}}(\boldsymbol{x}) = \gamma \boldsymbol{D}(\boldsymbol{x}) \boldsymbol{A}_R + (1 - \gamma) \boldsymbol{1} \boldsymbol{a}^T$. Following the same arguments of the case above we observe that $\Pr\left[\{X_0, X_1, \ldots, X_i\} \subseteq R\right] = \left(\boldsymbol{\pi}_R^T \boldsymbol{P}_R^{\mathrm{PR}}(\boldsymbol{x}) \boldsymbol{1}\right)^i$ by the same induction argument, where now $\boldsymbol{\pi}_R$ is the stationary distribution associated to the PageRank walk on $G_R$, which always exists. Therefore let $T_r(\boldsymbol{P}_R^{\mathrm{PR}}(\boldsymbol{x}))$ be the expected hitting time to reach a node in $B$ over walks following the transition matrix $\boldsymbol{P}_R^{\mathrm{PR}}(\boldsymbol{x})$ with the link-insertions provided by $\boldsymbol{x}$, we have

$$f_{\boldsymbol{\pi}_R}^{\mathrm{PR}} = \sum_{r \in R} \boldsymbol{\pi}_r T_r(\boldsymbol{P}_R^{\mathrm{PR}}(\boldsymbol{x})) = \sum_{k \geq 0} (\boldsymbol{\pi}_R^T \boldsymbol{P}_R^{\mathrm{PR}}(\boldsymbol{x}) \boldsymbol{1})^k$$
$$= (1 - \boldsymbol{\pi}_R^T (\gamma \boldsymbol{D}(\boldsymbol{x}) \boldsymbol{A}_R + (1 - \gamma) \boldsymbol{1} \boldsymbol{a}^T) \boldsymbol{1})^{-1}$$
$$= (1 - \gamma \boldsymbol{\pi}_R^T \boldsymbol{D}(\boldsymbol{x}) \boldsymbol{A}_R \boldsymbol{1} - (1 - \gamma) \boldsymbol{\pi}_R^T \boldsymbol{1} \boldsymbol{a}^T \boldsymbol{1})^{-1}$$
$$= \gamma^{-1} (1 - \boldsymbol{\pi}_R^T \boldsymbol{D}(\boldsymbol{x}) \boldsymbol{A}_R \boldsymbol{1})^{-1} \,,$$

---

[8] $\|\cdot\|_2$ denotes the $\ell^2$-norm (spectral norm).

by noting that $\boldsymbol{\pi}_R^T \mathbf{1} \boldsymbol{a}^T \mathbf{1} = 1$, which concludes the proof. $\qquad\square$

*Proof of Lemma 3.* We use a standard exchange argument. Let $\boldsymbol{x}^{\mathrm{G}}$ be the greedy solution returned by Algorithm algorithm 1, and suppose for contradiction that there exists $\boldsymbol{x}^* = (x_1^*, \ldots, x_{n_R}^*)$ with $p(\boldsymbol{x}^*) < p(\boldsymbol{x}^{\mathrm{G}})$. Without loss of generality, assume $\sum_i x_i^* = \sum_i x_i^G \leq b$. Let $i_1, i_2, \ldots, i_\ell$ (with $\ell \leq b$) be the red nodes chosen by the greedy algorithm in order, where $i_j$ is the node receiving an edge at iteration $j$ because it maximizes the marginal decrease.

We transform $\boldsymbol{x}^*$ into $\boldsymbol{x}^{\mathrm{G}}$ by ensuring that, for each $j = 1, \ldots, \ell$, the node $i_j$ in $\boldsymbol{x}^{\mathrm{G}}$ receives the same number of edges in $\boldsymbol{x}^*$. Concretely, at step $j$:

1. if $\boldsymbol{x}^*$ already allocates at least as many edges to $i_j$ as in the greedy solution up to iteration $j$, do nothing;
2. otherwise, pick any node $h \in R$ with $x_h^* \geq 1$ and $h \neq i_j$, remove one of its edges, and assign it to $i_j$.

Since at iteration $j$ we have $\Delta_{i_j} \geq \Delta_h$, such a reallocation cannot increase the objective. After $\ell$ steps, $\boldsymbol{x}^*$ matches $\boldsymbol{x}^{\mathrm{G}}$ exactly. Each reallocation preserves or decreases $p(\boldsymbol{x}^*)$, and thus we contradict the assumption $p(\boldsymbol{x}^*) < p(\boldsymbol{x}^{\mathrm{G}})$. $\qquad\square$

*Proof of Lemma 4.* Note that we can express $f_{\boldsymbol{\pi}}(\boldsymbol{x}) = \mathbb{E}_{r \sim \boldsymbol{\pi}_R}[T_r(\boldsymbol{P}(\boldsymbol{x}))] = \sum_{r \in R} \boldsymbol{\pi}_r T_r(\boldsymbol{P}(\boldsymbol{x}))$ and similarly $f_{\mathrm{avg}}(\boldsymbol{x}) = \mathbb{E}_{r \sim \boldsymbol{u}_R}[T_r(\boldsymbol{P}(\boldsymbol{x}))]$. Therefore

$$|f_{\boldsymbol{\pi}}(\boldsymbol{x}) - f_{\mathrm{avg}}(\boldsymbol{x})| = |\mathbb{E}_{r \sim \boldsymbol{\pi}_R}[T_r(\boldsymbol{P}(\boldsymbol{x}))] - \mathbb{E}_{r \sim \boldsymbol{u}_R}[T_r(\boldsymbol{P}(\boldsymbol{x}))]|$$

$$\leq \sum_{r \in R} T_r(\boldsymbol{P}(\boldsymbol{x})) \left| \boldsymbol{\pi}_{R,r} - \frac{1}{n_R} \right|$$

$$\leq 2 \max_{r \in R} \{T_r(\boldsymbol{P}(\boldsymbol{x}))\} d_{TV}(\boldsymbol{\pi}_R, \boldsymbol{u}_R) .$$

Where the first inequality uses the triangle inequality. The second inequality bounds each $T_r(\boldsymbol{x})$ with the maximum and uses the definition of total variation distance between two discrete distributions $\boldsymbol{z}_1, \boldsymbol{z}_2$ defined on a set $\mathcal{Z}$: $d_{TV}(\boldsymbol{z}_1, \boldsymbol{z}_2) = \frac{1}{2} \sum_{i \in \mathcal{Z}} |\boldsymbol{z}_1(i) - \boldsymbol{z}_2(i)|$. For *undirected and connected* graphs let $\bar{\delta}_{G_R}$ be the average of all node degrees $|\delta_{G_R}(i)|, i \in R$ and $m_R = |E_R|$ then we have,

$$2 d_{TV}(\boldsymbol{\pi}_R, \boldsymbol{u}_R) = \sum_{r \in R} \left| \frac{|\delta_{G_R}(r)|}{2m_R} - \frac{1}{n_R} \right|$$

$$= \sum_{r \in R} \left| \frac{1}{2m_r} (|\delta_{G_R}(r)| - \bar{\delta}_{G_R}) \right|$$

$$= \frac{1}{2m_r} \sum_{r \in R} \left| |\delta_{G_R}(r)| - \bar{\delta}_{G_R} \right| ,$$

where we used the fact that $\boldsymbol{\pi}_{R,r} = \frac{|\delta_{G_R}(r)|}{2m_R}$ and the handshaking lemma. Therefore,

$$|f_{\boldsymbol{\pi}}(\boldsymbol{x}) - f_{\mathrm{avg}}(\boldsymbol{x})| \leq \frac{\max_{r \in R} \{T_r(\boldsymbol{P}(\boldsymbol{x}))\}}{2m_r} \sum_{r \in R} \left| |\delta_{G_R}(r)| - \bar{\delta}_{G_R} \right| .$$

$\qquad\square$

*Proof of Lemma 5.* Let $\hat{p}(\boldsymbol{x}) \doteq \hat{\boldsymbol{\pi}}_R^T \boldsymbol{D}(\boldsymbol{x}) \boldsymbol{A}_R \mathbf{1}$ for any $\boldsymbol{x} \geq 0$. For any $\boldsymbol{x} \geq 0$, it holds that:

$$|\hat{p}(\boldsymbol{x}) - p(\boldsymbol{x})| \leq |\langle \hat{\boldsymbol{\pi}}_R - \boldsymbol{\pi}_R, \boldsymbol{D}(\boldsymbol{x}) \boldsymbol{A}_R \mathbf{1} \rangle|$$

$$\leq \|\hat{\boldsymbol{\pi}}_R - \boldsymbol{\pi}_R\|_2 \cdot \|\boldsymbol{D}(\boldsymbol{x}) \boldsymbol{A}_R \mathbf{1}\|_2$$

$$\leq \varepsilon \cdot \sqrt{n_R} .$$

Note that $\hat{\boldsymbol{x}}$ is an optimal solution for the objective $\hat{p}$. By using this fact and the above inequality, we can easily conclude that

$$p(\hat{\boldsymbol{x}}) \leq \hat{p}(\hat{\boldsymbol{x}}) + \varepsilon \sqrt{n_R} \leq \hat{p}(\boldsymbol{x}^*) + \varepsilon \sqrt{n_R} \leq p(\boldsymbol{x}^*) + 2\varepsilon \sqrt{n_R} .$$

$\qquad\square$

---

**Algorithm 1:** PARITY: optimal average hitting time minimizer under stationary transitions

---

**Input:** Graph $G$, transition matrix $\boldsymbol{P}$, budget $b$, upper-bound vector $\boldsymbol{c}$.
**Output:** $\boldsymbol{x} = \boldsymbol{x}^*$ optimal solution to S-HTMP.

1   $\boldsymbol{x} \leftarrow \boldsymbol{0} \in \mathbb{R}^{n_R}$; max-priority queue $PQ \leftarrow \emptyset$;
2   Compute the stationary distribution $\boldsymbol{\pi}_R$ of $G_R$ proportional to $\boldsymbol{P}_R$;
3   $(\boldsymbol{M}_{11}, \ldots, \boldsymbol{M}_{n_R n_R}) \leftarrow \operatorname{diag}(\boldsymbol{\pi}_R \boldsymbol{1}^T \boldsymbol{A}_R)$;
4   **for** $i \leftarrow 1$ *to* $n_R$ **do**
5     **if** $\boldsymbol{x}_i < \boldsymbol{c}_i$ **then**
6       $\Delta_i \leftarrow \frac{\boldsymbol{M}_{ii}}{|\delta_G^+(i)| + \boldsymbol{x}_i} - \frac{\boldsymbol{M}_{ii}}{|\delta_G^+(i)| + \boldsymbol{x}_i + 1}$;        `// Compute gain.`
7       $PQ.\text{insert}((\Delta_i, i))$;

8   **for** $t \leftarrow 1$ *to* $b$ **do**
9     **if** $PQ = \emptyset$ **then break**;
10     $(\Delta_i, i) \leftarrow PQ.\text{pop}()$;              `// Highest gain` $\Delta_i$.
11     $\boldsymbol{x}_i \leftarrow \boldsymbol{x}_i + 1$;
12     **if** $\boldsymbol{x}_i < \boldsymbol{c}_i$ **then**
13       $\Delta_i \leftarrow \frac{\boldsymbol{M}_{ii}}{|\delta_G^+(i)| + \boldsymbol{x}_i} - \frac{\boldsymbol{M}_{ii}}{|\delta_G^+(i)| + \boldsymbol{x}_i + 1}$;        `// Recompute gain.`
14       $PQ.\text{insert}((\Delta_i, i))$;

15   **return** $\boldsymbol{x}$;

---

## B   PERSONALIZATION VECTORS FOR PAGERANK

We now introduce three simple and useful settings for $\boldsymbol{a}^T$, that we also use for input to PARITY in Appendix D.6,

1. (*uniform*) $\boldsymbol{a}_1 = \frac{1}{n_R}\boldsymbol{1}$, which assigns equal importance to all nodes in $G_R$.
2. (*size-based*) $\boldsymbol{a}_2^T = \frac{1}{n_R}(\kappa(1)/n_R, \ldots, \kappa(n_R)/n_R)$, where $\kappa : R \mapsto \mathbb{N}_{>0}$ is a function that assigns the size of the (strongly) connected component containing $i \in [n_R]$. That is, $\boldsymbol{a}$ assigns higher (and equal) importance to nodes in larger strongly connected components over $G_R$.
3. (*size- and degree-based*) $\boldsymbol{a}_3^T = \frac{1}{n_R}\left(\frac{|\delta_{G_R}(1)|}{4m_R} + \frac{|\kappa(1)|}{2n_R}, \ldots, \frac{|\delta_{G_R}(n_R)|}{4m_R} + \frac{|\kappa(n_R)|}{2n_R}\right)$, where $|\delta_{G_R}(i)| = |\delta_{G_R}^+(i)| + |\delta_{G_R}^-(i)|$ for directed graphs and $i \in R$. That is, $\boldsymbol{a}$ assigns higher importance to nodes in $R$ having higher degree and belonging to larger strongly connected components over $G_R$.

## C   PSEUDOCODE – PARITY

We show the pseudocode of our algorithm PARITY in Algorithm 1, reporting in details all the steps discussed in Sec. 4.2.

## D   EXPERIMENTS

### D.1   DATASETS

A summary of the undirected graph datasets and key statistics is reported in Table 1. We refer to the original publication introducing those datasets for more details Haddadan et al. (2021). We note that these datasets are collected from different domains, in particular `Guns`, `Abortion`, `Sociology`, and `Politics` are collected from Wikipedia; `Math` and `Tech` from books categories on Amazon; and `PolBlog` captures connections between political blogs.

Table 1: Datasets used in the experimental evaluation. We report: $|V|$ and $|E|$ the number of nodes and edges; $|R|$ (resp. $|B|$) the number of red (resp. blue) nodes; $|E|_{R \leftrightarrow B}$ the existing number of edges from $R$ to $B$; $\kappa$ the number of connected components over $G_R$, and the number of nodes (of $R$) in the largest component under $\kappa_{\max}$. We also mark under "Dir" if we used the dataset in our experiments as directed (D), undirected (U), or both. When marked with both D and U the statistics in the table refer to the undirected graph. The suffix $-\text{MET}$ denotes partitions of $R$ and $B$ obtained with METIS, while $-\text{RND}$ denotes partitions obtained at random.

| Name | $|V|$ | $|E|$ | $|R|$ | $|B|$ | $|E|_{R \leftrightarrow B}$ | $\kappa$ | $\kappa_{\max}$ | Dir |
|------|------|------|------|------|------|------|------|------|
| Guns | 251 | 550 | 134 | 117 | 132 | 10 | 124 | U,D |
| Math | 171 | 287 | 160 | 11 | 16 | 3 | 158 | U,D |
| Tech | 88 | 146 | 25 | 63 | 56 | 14 | 7 | U,D |
| Abortion | 604 | 1 585 | 208 | 396 | 232 | 14 | 189 | U,D |
| PolBlog | 1 222 | 16 717 | 636 | 586 | 1 575 | 11 | 622 | U,D |
| Sociology | 3 236 | 8 745 | 648 | 2 588 | 430 | 30 | 615 | U,D |
| Politics | 20 460 | 116 094 | 10 339 | 10 121 | 26 672 | 81 | 10 257 | U |
| S-br-RND | 56 739 | 212 945 | 45 246 | 11 493 | 67 755 | 4 012 | 40 537 | U |
| S-br-MET | 56 739 | 212 945 | 28 369 | 28 370 | 18 178 | 18 | 28 263 | U |
| Dblp-RND | 317 080 | 1 049 866 | 253 682 | 63 398 | 335 385 | 13 849 | 228 970 | U |
| Dblp-MET | 317 080 | 1 049 866 | 158 540 | 158 540 | 54 388 | 21 | 158 239 | U |
| Twitter-RND | 68 413 | 1 685 152 | 54 737 | 13 676 | 410 554 | 2 416 | 51 621 | D |
| Gplus-RND | 69 501 | 9 168 660 | 55 565 | 13 936 | 250 9144 | 1 382 | 54 111 | D |

To further test the scalability of our algorithm PARITY we also select four large datasets from widely used graph repositories.[9] Given that for such datasets we do not have ground truth labels for the nodes, we obtain the sets $R$ and $B$ as follows: ($i$) assigning randomly 80% of the nodes to $R$ and the rest of the nodes to $B$; ($ii$) using METIS (Karypis & Kumar, 1998) to partition the nodeset $V$ in two sets. Where METIS is a widely adopted scalable algorithm to compute balanced partitions of a graph $G$, in terms of nodes. The two resulting partitions are denoted, respectively with the suffix $-\text{RND}$ and $-\text{MET}$ in Table 1. Note that METIS can only be used on undirected graphs.

We use all datasets up to Sociology in the experiments of Sec. 5.1, while the last four datasets are used to assess PARITY's scalability in Appendix D.6. All our baselines cannot process Politics within two hours of runtime limit.

## D.2 ADDITIONAL BASELINES

Following previous work (Haddadan et al., 2021; Adriaens et al., 2023), we also considered the following baselines: (3) PR is a simple approach that selects nodes in $R$, likely to have large hitting times,[10] at random and connects them to nodes in $B$ if possible; (4) ROV (Garimella et al., 2017) outputs $k$ edges to reduce a controversy score defined over the groups red $R$ and $B$ (see Garimella et al. (2017); Haddadan et al. (2021) and Sec. 6 for more details); we also consider: (5) N-RCN and (6) N-RWCN, which are variations of RepBubLik that adopt simple greedy approaches to determine where to add new edges. Note that we do not consider other approaches, e.g., based on machine-learning techniques as they generally do not offer guarantees and are designed for simpler non-integer objective functions.

## D.3 EXPERIMENTAL ENVIROENMENT

We implemented PARITY in Python 3.8.0 and executed it on a server with 96 CPUs (4x Intel Xeon Gold 6252N CPU @ 2.30/3.60GHz) and 3TB of RAM. All the software was executed within a container running a Ubuntu image.[11]

---

[9] S-br and DBLP from https://networkrepository.com/ and Twitter and Gplus from https://snap.stanford.edu/data/

[10] These are named *parochial* nodes in the work of Haddadan et al. (2021).

[11] https://apptainer.org/

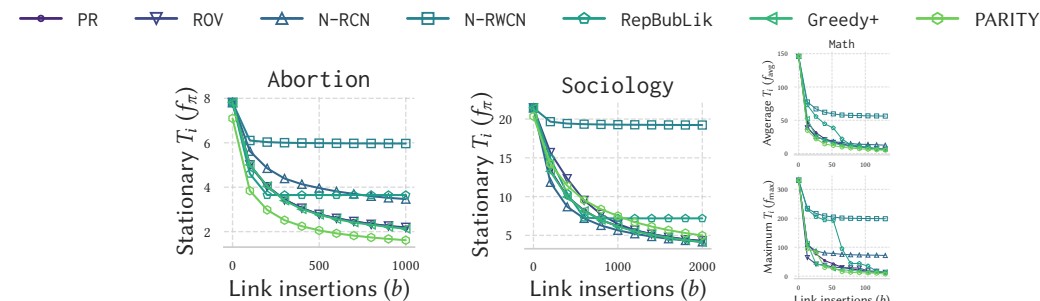

Figure 4: The first two figures report the results for the S-HTMP with budget $b$ missing from Sec. 5.1. The third figure reports missing results for the datasets in Table 1 under the setting of Sec. 5.1.

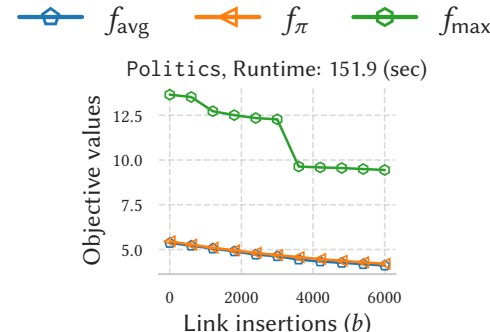

Figure 5: Results on the `Politics` data, PARITY is the only method completing its executions within two hours.

All experiments were performed single threaded, with a maximum RAM memory of 20GGB, with the exception of the experiments for large scale data in Appendix D.6, where we set 100GB of RAM memory limit. We considered all graphs as undirected given that the state-of-the-art approach by Adriaens et al. (2023) cannot be executed on directed graphs as it is impractical, unless otherwise stated.[12] For all settings, the time-limit of each execution was set to 2 hours, with the exception of the the the large-scale experiments of Appendix D.6 for which we allow 6 hours of computation. Details on the parameter settings are reported in Appendix D.4.

### D.4 PARAMETERS

The various baselines that we consider have various parameters, our setting is the same as the one considered by Adriaens et al. (2023). In particular for `RepBubLik` the parameters are set to $t = 15$ (corresponding to the length of the random walks), $b = 2$ (threshold for defining the good nodes), $k = 10$ (its percentage defining the nodes to consider), $m = 50$ (maximum edges to add to the graph). For the algorithm of `Greedy+` we use all the paramters as default, and we also note that some of them are hard-coded by the authors, i.e., the parameters defining the approximation to speed-up the method. Finally all the variants of `RepBubLik` consider $k = 10$, for the top-$k$ nodes on which to add the candidate edges. For more details refer to our source code (https://anonymous.4open.science/r/optimalHTM) and the source code of Adriaens et al. (2023).

Regarding PARITY$_{\hookrightarrow}^{PM}$ we set the number of iterations of the power method to 30.

---

[12]https://github.com/HonglianWang/Minimizing-Hitting-Time-between-Disparate-Groups-with-Shor tree/main

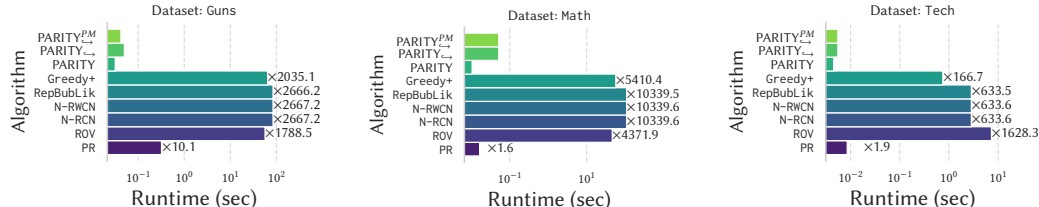

Figure 6: Missing results from Sec. 5.2. Runtime of all algorithms to process the highest value of $b$ in the setting of Figure 2. We report, for each dataset the runtime of PARITY when executed on the respective directed graph and coupled with an exact computation of the stationary distribution $\boldsymbol{\pi}$ (marked with PARITY$_{\hookrightarrow}$) or with the power-method to approximate the stationary distribution (marked with PARITY$_{\hookrightarrow}^{PM}$). We also report the speedup of PARITY over each baseline algorithm on undirected graphs.

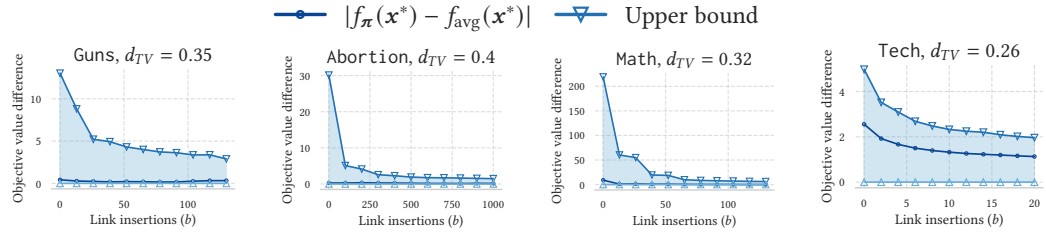

Figure 7: For each dataset we report the total variation distance $d_{TV}(\boldsymbol{\pi}_R, \boldsymbol{u}_R)$ between the stationary distribution over $G_R$ and the uniform distribution $\boldsymbol{u}_R$. Each figure shows the objective value difference $|f_{\boldsymbol{\pi}}(\boldsymbol{x}^*) - f_{avg}(\boldsymbol{x}^*)|$ where $\boldsymbol{x}^*$ is the solution reported by PARITY for the S-HTMP. In addition, we also report the upper bound $(2 \max_r\{T_r(\boldsymbol{P}(\boldsymbol{x}^*))\}d_{TV}(\boldsymbol{\pi}_R, \boldsymbol{u}_R))$ for the objective value difference as obtained from Lemma 4.

## D.5 BASELINE EVALUATION FOR THE S-HTMP

To assess the objective function $f_{\boldsymbol{\pi}}(\boldsymbol{x})$ (in Sec. 5.1) for the baselines we considered we proceeded as follows.

- We execute each baseline algorithm with budget $b$ to obtain an link-insertion vector $\boldsymbol{x}_{BL}$ where $BL$ refers to each baseline algorithm.
- we then considered the objective function $f_{\boldsymbol{\pi}}(\boldsymbol{x}_{BL})$ and evaluated its objective, computing a matrix inversion.

## D.6 ADDITIONAL RESULTS

**Missing results.** We report the results on missing datasets from Table 1 (see Appendix D.1) under the setting of Sec. 5.1 in Figures 4 to 6.

**Tightness of Lemma 4.** For ease of notation let $\Delta(f_{\boldsymbol{\pi}}, f_{avg}, \boldsymbol{x}^*) \doteq |f_{\boldsymbol{\pi}}(\boldsymbol{x}^*) - f_{avg}(\boldsymbol{x}^*)|$ for $\boldsymbol{x}^*$ computed by PARITY. To assess the tightness of Lemma 4 that quantifies the deviation $\Delta(f_{\boldsymbol{\pi}}, f_{avg}, \boldsymbol{x}^*)$ we empirically evaluated the upper bound $2 \max_{r \in R}\{T_r(\boldsymbol{P}(\boldsymbol{x}^*))\}d_{TV}(\boldsymbol{\pi}_R, \boldsymbol{u}_R)$ under the setting of Sec. 5.1 for **Q2** on undirected graphs. We show the value of $d_{TV}(\boldsymbol{\pi}_R, \boldsymbol{u}_R)$ and the tightness of the bound of Lemma 4 in Figure 7. We first note that in practice $\Delta(f_{\boldsymbol{\pi}}, f_{avg}, \boldsymbol{x}^*)$ tends to be very small, approaching 0 on most datasets and different values of $b$. However, we note that for small values of $b$ the upper-bound provided by Lemma 4 tends to be quite large on most configurations (e.g., Guns, Abortion and Math), but Tech. That is, the bound from Lemma 4 is often quite conservative, due to two main factors: (i) the high total variation distance $d_{TV}(\boldsymbol{\pi}_R, \boldsymbol{u}_R)$ between the stationary distribution and the uniform distribution; and (ii) the fact that on most configurations the value $\max_{r \in R}\{T_r(\boldsymbol{P}(\boldsymbol{x}^*))\}$ remains high. That is, reducing the maximum hitting time from nodes in $R$ to those in $B$ is extremely challenging, especially for small budget $b$ (see Sec. 5.1).

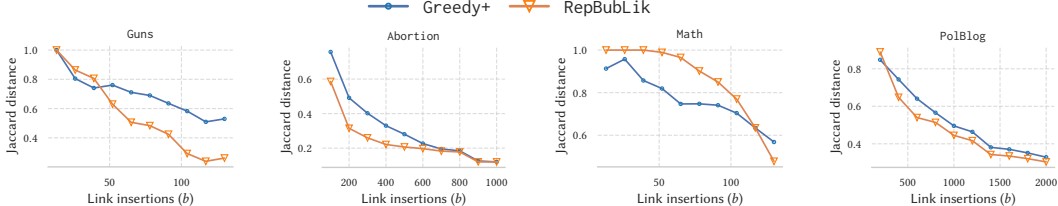

Figure 8: Jaccard distance between the set of nodes $i \in R$ selected by PARITY for new link insertions of the form $(i, b)$ with $b \in B$, and the set of nodes for new link insertions $i \in R$ selected by the baseline algorithms.

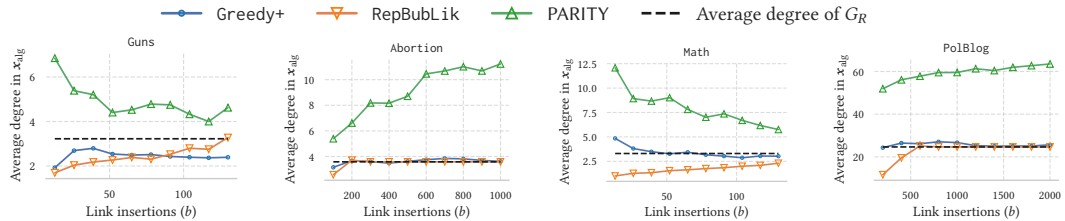

Figure 9: Average degree distribution over the multiset of nodes $i \in R$ in the solution $\boldsymbol{x}_{\mathrm{alg}}$ provided by the different methods ($\mathrm{alg} \in \{\mathsf{PARITY}, \texttt{Greedy+}, \texttt{RepBubLik}\}$).

As a summary, While in practice $\Delta(f_{\boldsymbol{\pi}}, f_{\mathrm{avg}}, \boldsymbol{x}^*)$ tends to be very small, and close to 0, our theoretical result cannot fully characterize such behavior. We leave as an open question, identifying tighter bounds for $\Delta(f_{\boldsymbol{\pi}}, f_{\mathrm{avg}}, \boldsymbol{x}^*)$, possibly independent of $\max_{r \in R}\{T_r(\boldsymbol{P}(\boldsymbol{x}^*))\}$.

**Solution comparison.** In this section we now compare the solutions reported by PARITY, `Greedy+`, and `RepBubLik` under the setting of Sec. 5.1, where all graphs are undirected. We select baselines `Greedy+`, and `RepBubLik`, given that they achieve comparable performances to PARITY for $f_{\mathrm{avg}}$ and $f_{\mathrm{max}}$. In particular, we first assess the difference among the solutions $\boldsymbol{x}_{\texttt{Greedy+}}$ and $\boldsymbol{x}_{\texttt{RepBubLik}}$ and the solution reported by PARITY. Let $R_{\mathrm{alg}} = \{i \in R : \boldsymbol{x}_{\mathrm{alg}} > 0\}$ for $\mathrm{alg} \in \{\mathsf{PARITY}, \texttt{Greedy+}, \texttt{RepBubLik}\}$ be the *set* of nodes for which there exists at least one new link of the form $(i, b), i \in R, b \in B$ to be added to $G$ according to the solution provided by an algorithm alg. In Figure 8 we display the Jaccard-distance $d_J(R_{\mathsf{PARITY}}, R_{\texttt{Greedy+}})$ ($d_{J,1}$ for short) and $d_J(R_{\mathsf{PARITY}}, R_{\texttt{RepBubLik}})$ ($d_{J,2}$ for short); for varying budget $b$. We recall that for two non-empty sets $A, B$ it holds that

$$d_J(A, B) = \frac{|A \cup B| - |A \cap B|}{|A \cup B|} \ ,$$

where $d_J(A, B)$ ranges from 0 and 1, approaching 0 where $A$ and $B$ share most of their elements and 1 when the two sets are disjoint. We observe that the nodes $i \in R_{\mathsf{PARITY}}$ differ significantly by both the set of nodes $i \in R$ selected by `Greedy+` and `RepBubLik` on most datasets. That is, both distances $d_{J,1}$ and $d_{J,2}$ are above 0.5 on most configuration, with PARITY having a very different strategy for its new link additions especially when the budget $b$ is small compared to $|R|$. Clearly, with increasing $b$ the Jaccard distances $d_{J,1}$ and $d_{J,2}$ are expected to decrease. That is, with large enough $b$ all the nodes $i \in R$ not already connected to all nodes in $B$ will have at least one new link insertions of the form $(i, b)$ for some $b \in B$.

Note that the Jaccard distance in Figure 8 does not take into account the fact that multiple edges of the form $(i, b_1), (i, b_2), \ldots$ can be selected by the various algorithms. Therefore, to further study the difference in the solutions provided by the different approaches we compute the average node degree in $G_R$ over the multiset[13] of nodes selected for new link additions by the various methods. The results are displayed in Figure 9. We observe that the average degree $|\delta_R(i)|$ of the nodes $i \in R$ selected by PARITY is significantly higher compared to the average degree of the nodes $i \in R$

---

[13] A multiset allows duplicate elements.

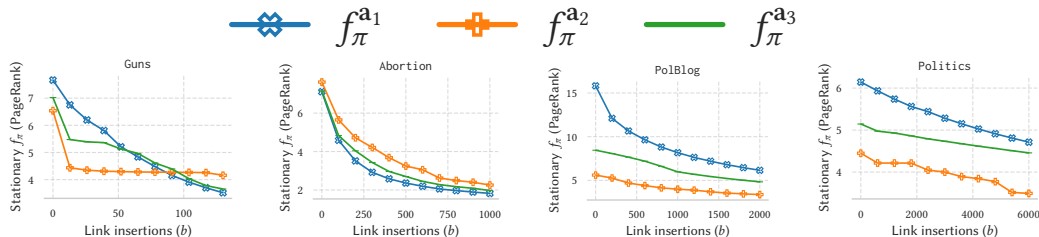

Figure 10: Objective value for $f_{\boldsymbol{\pi}}$ under $\boldsymbol{P} = \boldsymbol{P}^{\mathrm{PR}}$ and different personalization vectors after $b$ link insertions. We mark each different objective according to the personalization $\boldsymbol{a}_i$ as described in Appendix B.

Table 2: Runtime (in seconds) of PARITY on our largest datasets to optimize $f_{\boldsymbol{\pi}}$ under the setting of Sec. 5.1. Following the ordering of the datasets, we set $b$ as $40K$, $30K$, $1K$ and $10K$.

| S-br-RND | S-br-MET | Dblp-RND | Dblp-MET | Twitter-RND | Gplus-RND |
|---|---|---|---|---|---|
| 564.8 | 1 017.2 | 21 935.5 | 57 876.6 | 1 175.1 | 1 133.6 |

selected by baselines Greedy+ and RepBubLik, and also significantly higher than the average degree over $G_R$. Such behavior is not surprising, recall that our new objective assigns more importance to nodes $i \in R$ with higher value $\boldsymbol{\pi}_{R,i}$ over the stationary distribution $\boldsymbol{\pi}$. Furthermore, recall that for undirected graphs $\boldsymbol{\pi}_{R,i} \propto |\delta_R(i)|$, showing that effectively our new problem formulation encourages more important nodes over $G_R$ to be selected for new link additions. Interestingly, as captured by our theoretical result in Equation (5), PARITY does *not* simply select nodes with higher degree $|\delta_R(i)|$. For example, on the Abortion and PolBlog datasets the average degree in the solution provided by PARITY increases with with higher value of $b$, highlighting that nodes with the highest degree are selected only when $b$ is large enough. That is, the optimal solution to Problem 2 will contain link insertions of the form $(i, b)$ carefully balancing the importance of nodes $i \in R$ over $G_R$ through $\boldsymbol{\pi}_{R,i}$, combined with the connectivity of nodes $i \in R$ over $G$ captured by $|\delta_G^+(i)|$, a non-trivial result.

**Scalability on large data.** To test PARITY on large datasets we considered the four largest datasets of Table 1, processed according to Appendix D.1. We show the runtime of PARITY to optimize each network in Table 2. Note that for directed networks (see Table 1) we use PARITY coupled with 30 iterations of the power-method. Unfortunately, on such large datasets we cannot afford evaluating the objectives $f_{\boldsymbol{\pi}}$ due to extremely expensive cost of computing a large matrix inversion, therefore we only report PARITY's runtime. As we can observe, PARITY can process large datasets very efficiently, i.e., most of the datasets are processed in less than 20 minutes, with the exception of Dblp (both -RND and -MET) which requires slightly more than 4 hours. We note that the time to process the graph obtained through METIS is about a factor 2 higher than the graph obtained with random partitions, for both datasets S-br and Dblp. This difference can be partially explained by the fact that there are fewer edges of type $R \times B$ in the partitions obtained with METIS, see Table 1. Note that the Dblp dataset has more than $300K$ nodes: its size is more than $10\times$ larger compared to Politics. Recall, that Politics cannot be processed within two hours of computation with current methods. We note the large improvement over existing methods brought by PARITY. That is, under the speedup of Sec. 5.2, the expected time to process the Dblp dataset for our baselines is more than 40 days of computation.

**PageRank walks.** Here we answer **Q4**. In this setting we considered several datasets of Table 1, and used PARITY to optimize $f_{\boldsymbol{\pi}}$ under $\boldsymbol{P} = \boldsymbol{P}^{\mathrm{PR}}$, for the three different personalization vectors introduced in Appendix B. We set the damping factor as $\gamma = 0.85$, and we consider each dataset as directed. To compute the stationary distribution under $\boldsymbol{P}^{\mathrm{PR}}$ we use 30 iterations of the power-method. We remark, that since the stationary distribution of PageRank walks is guaranteed to exist, we compute $f_{\boldsymbol{\pi}}$ on the entire graph $G_R$. That is, we do not focus on the largest connected component as in Sec. 5. We do not report the runtimes since required by PARITY since it is identical

to PARITY$_{\hookrightarrow}^{PM}$ from Sec. 5.2 (up to a 5% of variability on the respective data), for `Politics` the runtime refers to Figure 5.

We observe that, as expected, different personalization lead to different objectives for the S-HTMP. In particular, on some datasets, there could be significant variability (e.g., `PolBlog` and `Politics`) where $f_{\boldsymbol{\pi}}^{\boldsymbol{a}_2}$ and $f_{\boldsymbol{\pi}}^{\boldsymbol{a}_3}$ differ significantly, in constrast with the `Abortion` where all personalizations lead to similar objectives. Such variability may depend on different dataset properties over $G_R$, e.g., the density of the various connected components.

**Summary for Q4.** Overall our experiments show that PARITY is extremely versatile, as it can minimize the S-HTMP under various personalizations for PageRank-based walks over $G$, according to the application of interest.

