# OpenReview forum: "Optimal and Efficient Link Insertion for Hitting-Time Minimization"
_ICLR.cc/2026/Conference — Submitted to ICLR 2026_

### Official Review · Reviewer_fDD5 · 2025-10-20

**Soundness:** 3
**Presentation:** 3
**Contribution:** 2
**Rating:** 2
**Confidence:** 4

**Summary:**

This paper studies the problem of minimizing hitting times between two node groups in a graph via link insertion. The study focuses on a specific case where the objective is a weighted sum of hitting times, with weights given by the random-walk stationary distribution of the source subgraph. For this scenario, the authors theoretically establish that the problem admits a closed-form objective whose optimal solution can be computed efficiently with a simple greedy algorithm. Empirical results demonstrate the proposed method's superior efficiency and solution quality under various metrics.

**Strengths:**

1. The problem of minimizing hitting time via link insertion is interesting and meaningful with a clean mathematical formulation. Also, the idea of assigning weights according to the random-walk stationary distribution is intuitive.
2. The insights behind deriving mathematical properties and an optimal algorithm for the stationary hitting-time minimization problem are clear and elegant in general.
3. Thorough experiments are conducted, and in particular the quality of the proposed method is evaluated using average and maximum hitting times.

**Weaknesses:**

1. A major concern is that this paper may not align closely with the interests of the ICLR community. The studied problem is essentially a combinatorial optimization problem in network science, and the proposed method is a classical greedy algorithm without using deep learning techniques. Thus, it seems that the work is out of scope for ICLR (cf. relevant topics listed in https://iclr.cc/Conferences/2026). Worse still, potential applications and connections to deep learning are not discussed, and none of the references are from machine learning venues. The paper seems more suitable for data mining or network science venues.
2. The technical depth is relatively limited. The work introduces the special case where the weights in the objective come from the random-walk stationary distribution, which greatly simplifies the problem. It is not surprising that efficient algorithms exist for this special case. Specifically, the analysis in this case seems somewhat trivial given the observations from previous work and the property that the probability distribution in the subgraph remains the stationary distribution before hitting the other subgraph, yielding a closed form of the objective function. Once this is done, the greedy algorithm for minimizing this objective is immediate.
3. While the experiments show that the proposed algorithm achieves remarkable solution quality in terms of both the average and maximum hitting times, these appear as heuristic results to me and thus limit the overall contribution of the paper. The approach can be interpreted as optimizing the proposed objective $f_{\pi}$​ and hoping that the solution also performs well for $f_{\mathrm{avg}}$​ and $f_{\max}$​. However, no theoretical guarantees or intuitive explanations are provided to justify this transfer of performance. In fact, intuitively, minimizing $f_{\pi}$​ and $f_{\max}$ seem to be two very different tasks, as the former puts more weight on "important" nodes in $R$ and the latter aims at minimizing the hitting time from the "least important" node in $R$, at least in some sense. This intuition is also reflected by the fact that the $f_{\max}$​ value achieved by `PARITY` indeed remains large on the `PolBlog` dataset.
4. The paper should go through further proofreading to correct several minor issues. See the minor comments below for detailed examples.

Minor comments:
1. Please correct the hyphenation of "hitting time" for consistency. It should be hyphenated only when used as a compound adjective (e.g., in the title); when used as a noun phrase (e.g., Line 12: "...the hitting time between..."), the hyphen should be omitted.
2. Lines 18-20: the sentence is confusing to me if I do not read the whole paper. "The problem is generally NP-hard" and "when the random walk follows stationary transitions" seem inexact and hard to understand.
3. Line 43: I recommend adding a reference after "... is NP-hard"
4. Line 108: $T_{i}$ -> $T_{i}(\boldsymbol{P})$
5. Line 123: it is better to introduce the notations $\boldsymbol{e}_{i}$ and $\boldsymbol{1}$
6. Line 138: $\boldsymbol{x}\_{i} \leq \boldsymbol{c}\_{i}$ -> $\boldsymbol{c}\_{i}$
7. Line 182: it should be "the random walk on $G_{R}$ follows $\boldsymbol{P}$"?
8. Line 291: add a comma after "Summarizing"
9. Lines 456-458: this should be one sentence
10. Line 459: remove the comma
11. Line 613: it is somewhat weird to use $\boldsymbol{\pi}\_{R,r}$ to denote the $r$-th entry in $\boldsymbol{\pi}\_{R}$

**Questions:**

1. Can you provide discussions and references to show that this work is interesting to the ICLR community?
2. In Line 252, I think it only holds that $\boldsymbol{\pi}\_{R}^{\top}\boldsymbol{D}(\boldsymbol{x})\boldsymbol{A}\_{R}^{\top}\boldsymbol{1} = \langle \boldsymbol{M},\boldsymbol{D}(\boldsymbol{x}) \rangle\_{F}$. Did I miss something, or should $\boldsymbol{M}$ be defined to involve $\boldsymbol{A}_{R}^{\top}$ instead?
3. Can you provide intuitive explanations as to why `PARITY` can also achieve small $f_{\mathrm{avg}}$​ and $f_{\max}$ (under some assumptions)?

---

> ### Author Response · Authors · 2025-11-24
>
> We thank the reviewer for the very detailed and constructive feedback. Below we answer the questions.
>
> 1.
> We answer this question in our common response and in the revised manuscript.
>
> 2.
> We thank the reviewer for pointing out the typo in the definition of $\mathbf{M}$.
> We fixed the typo as $\mathbf{M} = \pi_R \mathbf{1}^T \mathbf{A}_{R}^T$, and in subsequent appeareances.
>
> Note that we use matrix $ \mathbf{M} $ only for presentation purposes.
> In our algorithm we directly compute $ \mathbf{M}_{ii} $ in linear time $O(n_R + m_R)$
>
> 3.
> In our revised version we provide a new lemma relating the accuracy over $f_{\text{avg}}$ and $f_{\mathbf{\pi}}$ for the solution of PARITY. Let $\mathbf{x}^\*$ be the solution identified by PARITY. Our new lemma provides a bound on the distance $|f_{\text{avg}}(\mathbf{x}^\*) - f_{\mathbf{\pi}}(\mathbf{x}^\*)|$, which depends on two key quantities:
>
> A. the total-variation distance between $\mathbf{\pi}_R$ and the uniform distribution over $n_R$.
>
> B. the maximum hitting time of a node $r\in R$ under the edge additions $\mathbf{x}^\*$, which can be further bounded with $O(|R|^{3/4} f_{\text{avg}}(\mathbf{x}^\*))$ using results from prior works.
>
> Hence, if the stationary distribution $\pi$ is close in total-variation distance to the uniform weighting then PARITY achieves a close-to-optimal solution for the average objective.
> Unfortunately, obtaining a non-trivial connection with the maximum objective appears to be significantly more challenging, and therefore we do not have a rigorous result about the quality of our solution for $f_{\text{max}}$. We think that this is a very interesting and challenging problem for future work.
>
> We thank the reviewer for pointing out various minor comments
> 1. We fixed the use of "hitting-time" vs "hitting time";
> 3. We added the references proving, or mentioning similar NP-Hard objectives;
> 7. We use $\mathbf{\pi}_R$ to denote that the walk follows stationary transitions over $G_R$;
> 11. We understand the concern, however we use $\mathbf{\pi}_R,r$ to avoid a messy double subscript notation;
> We fixed the typos and notation in points 4. 5. 6. 8. 9. 10.

---

> > ### Comment · Reviewer_fDD5 · 2025-11-25
> >
> > Thanks for the reply. I still have the following concerns.
> >
> > 1. While the added related works for machine learning make sense to me, I believe that the authors should try to present them as a motivation at the beginning of the paper to make this work more interesting and accessible to the ICLR community.
> > 2. It seems to me that the phrasing "the random walk follows the stationary distribution/transitions" that appears throughout (e.g., Lines 19 and 184) sounds unusual. The correct one should be "the random walk starts from the stationary distribution and follows the transition matrix." The two points stated in Line 186 also essentially boil down to a single point (ii). Please let me know if the misunderstanding is on my side.
> > 3. Regarding the added Lemma 4, first, the authors should briefly introduce the notion of total variation distance and check whether $T_r(\boldsymbol{x})$ should be $T_r(\boldsymbol{P}(\boldsymbol{x}))$ according to previous notations. Second, I recommend adding discussions based on Lemma 4 to the experimental section, like verifying that the two quantities in Lemma 4 are indeed small in practice. Also, as you mentioned an upper bound on the maximum hitting time of a node under the edge additions in your Official Comment, it would be better to include it in the paper.
> > 4. I would like to know whether the authors have a response to Weakness 2 in my review.
> >
> > Please also consider the following minor comments.
> > 1. Line 290: remove the extra comma after $d_{TV}(\cdot)$
> > 2. Line 780: remove the redundant parentheses around the absolute value and add punctuation around this displayed math
> > 3. Line 1004: asses -> assess

---

> > > ### Author Response · Authors · 2025-11-28
> > >
> > > We thank again the reviewer for the very constructive feedback. We now answer the concerns.
> > >
> > > 1. To strengthen the possible interest of the ICLR community towards our work we now:
> > > - mention possible applications of our problem setting for machine learning tasks in the abstract and introduction;
> > > - mention the application of our methods for machine learning tasks in the conclusion section.
> > > 2.
> > > - We fixed the statement "follows the stationary distribution" with "starts from the stationary distribution" in both occurrences (lines 19 and 184 of the previous version of the manuscript). We agree with the reviewer that the new sentence is a more standard phrasing.
> > > - We agree with the reviewer that (i) + (ii) can be unified, focusing on (ii). We rephrased the entire paragraph in our updated version.
> > > 3.
> > > - We fixed the notation $T_r(\mathbf{P(x)})$ and introduced the notion of total variation distance.
> > > - We added new experimental results validating the tightness of the bound in Lemma 4 in Appendix D.6. As a summary of our results, the upper bound is generally loose in practice, but the distance $|f_{\pi}(\mathbf{x}^\*)-f_{\mathrm{avg}}(\mathbf{x}^\*)|$ is often very close to 0.
> > > - We mentioned the upper bound on $\max_{r\in R}T_r(\mathbf{P(x)})$ from previous work as stated in our Official Comment. We also mention that the bound cannot be used to prove an approximation ratio to the optimal solution of $f_{\text{avg}}$.
> > > 4. Quoting from the Official Review
> > > > The technical depth is relatively limited. The work introduces the special case where the weights in the objective come from the random-walk stationary distribution, which greatly simplifies the problem. It is not surprising that efficient algorithms exist for this special case. Specifically, the analysis in this case seems somewhat trivial given the observations from previous work and the property that the probability distribution in the subgraph remains the stationary distribution before hitting the other subgraph, yielding a closed form of the objective function. Once this is done, the greedy algorithm for minimizing this objective is immediate.
> > >
> > >    We agree with the reviewer that our use of stationary weights  substantially simplifies the HTMP problem.  This is, in fact, at the core of our contribution.  Our simpler but highly _non-trivial_ problem formulation enables the efficient and scalable optimization of the HTMP. The resulting algorithm (PARITY) is conceptually simple and easy to implement. In addition, our technically rigorous analysis leverages properties of the stationary distribution and of the HTMP objective. Furthermore, we avoid relying on complex and computationally heavy approaches from optimization theory.  Importantly, our approach applies to directed graphs, which are common in real-world applications and substantially more challenging than undirected graphs. We also mention that our work shows that there exist classes of graphs for which the HTMP can be solved in polynomial time.
> > >    Summarizing, we believe that the combination of (i) a meaningful and non-trivial problem formulation, (ii) a rigorous analysis, and (iii) an efficient algorithmic solution makes our contribution relevant to both practitioners and theorists.
> > >
> > > We addressed all minor comments mentioned by the reviewer.
> > >
> > > Together with our response we also updated our revised manuscript, implementing the modifications detailed above marked in blue.

---

### Official Review · Reviewer_mNfx · 2025-10-26

**Soundness:** 3
**Presentation:** 2
**Contribution:** 3
**Rating:** 6
**Confidence:** 2

**Summary:**

The paper studies how to efficiently reduce the expected time it takes for a random walk to travel from one group of nodes to another in a network by adding a limited number of new links. The classical version of this problem is known to be computationally hard. To address this, the authors introduce a new formulation that assigns different importance weights to nodes based on their stationary probabilities in the network. This modification makes the problem mathematically tractable and leads to a simple greedy algorithm called PARITY, which the authors prove to be globally optimal under their new setting. Experiments on several real-world and synthetic datasets show that PARITY achieves lower average hitting times and much faster runtime than existing baselines.

**Strengths:**

1. The paper provides a clear and correct mathematical reformulation of a known hard problem, turning it into one that can be solved optimally in polynomial time. The reasoning chain is coherent and rigorous.

2. The key idea of introducing node importance through stationary weights is original in this context and central to the proposed method’s efficiency and solvability. The proposed greedy procedure is straightforward to implement, easy to interpret, and mathematically proven to yield the optimal solution for the reformulated problem.

3. Experiments demonstrate very large speedups compared with existing heuristics and confirm that the method scales to networks with millions of edges.

**Weaknesses:**

1. The authors solve an easier, modified version of the problem by introducing node weights. This is fine, but the paper sometimes gives the impression that it solved the original NP-hard version. The distinction should be made clearer.

2. For several of the larger datasets, the groups are created by random assignment. Although this is a test for the algorithm's scalability, it divorces the experiment from the paper's core motivation, where groups typically represent pre-existing, meaningful partitions.

3. To fully realize the potential of this work for applications like polarization reduction, it would be valuable to include an analysis that correlates the reduction in hitting time with an established measure of network polarization.

**Questions:**

Please see the weaknesses above, and I expect the authors to address them in responses/rebuttals.

---

> ### Author Response · Authors · 2025-11-24
>
> We thank the reviewer for the positive comments. Below we address the concerns.
>
> 1. In the revised manuscript we made the distinction more clear, adding the following sentence.
>
> > In contrast to the general problem HTMP stated to be NP-hard, our new problem variant named S-HTMP is computationally tractable.
>
> 2. As pointed by the reviewer we only use random partitions for scalability tests. Unfortunately, we did not find large labelled datasets with semantic meaning for the partitions $R$ and $B$.
> Note that the largest dataset tested by the state-of-the-art work (Adriaens et al.) contained no more than 4000 nodes and 9000 edges. In this regard, our algorithm scales significantly beyond existing methods.
>
> 3. Our goal is to solve the S-HTMP problem in its general formulation, which is motivated by the study of polarization in graphs. We do agree that there are several potential scores that capture polarization that can be expressed through a function of hitting times. However, one of the advantages of our method PARITY is that it does not focus directly on any specific polarization score.
> This makes our algorithm of potential interest for various other applications, including deep learning applications requiring the minimization of commute times by $k$ edge additions (see our common response).

---

> > ### Comment · Reviewer_mNfx · 2025-11-27
> >
> > Thanks for the detailed responses. I still have concerns regarding my Weakness 2, but I will keep the original score.

---

> > > ### Author Response · Authors · 2025-12-03
> > >
> > > We thank the reviewer for the comment. We now address the concerns.
> > >
> > > Quoting from the Official Review, Weakness 2 corresponds to
> > > > For several of the larger datasets, the groups are created by random assignment. Although this is a test for the algorithm's scalability, it divorces the experiment from the paper's core motivation, where groups typically represent pre-existing, meaningful partitions.
> > >
> > > To address this issue in our revised version:
> > >
> > > We supplement our already extensive experimental evaluation with additional scalability tests.
> > > In our new experiments, we test the performances of PARITY on graphs partitioned through the METIS algorithm. (see App. D.6. paragraph: Scalability on large data).
> > > In summary, PARITY can efficiently process very large graphs partitioned with METIS, confirming the results obtained through random partitions.

---

### Official Review · Reviewer_JVvJ · 2025-10-29

**Soundness:** 3
**Presentation:** 4
**Contribution:** 3
**Rating:** 6
**Confidence:** 4

**Summary:**

This paper introduces the stationary hitting-time minimization problem (S-HTMP), which aims to reduce the expected steps for a random walk to travel from one node group to another by adding a limited number of edges. Unlike prior work, it weights nodes by their stationary distribution in the source subgraph, reflecting their structural importance.

The key contribution is PARITY, an optimal and efficient greedy algorithm that solves S-HTMP in polynomial time for both directed and undirected graphs. It achieves up to “10,000× speedup” over state-of-the-art methods, scales to million-edge graphs, and performs well even on traditional hitting-time metrics.

**Strengths:**

Novelty: This article redefines the established hit time minimization problem (HTMP) by introducing weighted objectives, where node importance is defined by the stationary distribution of random walks on the source subgraph (S-HTMP). This is a conceptual shift towards standardized weighting. Although using a stationary distribution is not a new concept, applying it to finding the optimal solution for a computational problem (HTMP) in polynomial time is a relatively novel idea.
Quality: The article provides rigorous theoretical arguments, including key lemmas for simplifying the objective function when using stationary distributions and optimality proofs for greedy parity algorithms. The experimental evaluation is comprehensive and convincing, demonstrating tremendous acceleration (up to 10000 times), scalability, and effectiveness.
Clarity: The paper is well-organized and presented clearly.
Contribution: From a theoretical perspective, it identifies an optimal point in the problem space - the S-HTMP formula - where previously tricky problems become manageable, providing a new perspective for graph modification problems and a highly scalable and efficient tool for tasks with real-world impact.

**Weaknesses:**

1. The paper did not fully discuss and compare why the stationary distribution πR was chosen as the node weighting scheme. It is currently unclear why πR is a more meaningful node importance measure than other impact measures.
2. Incomplete validation of the core HTMP: Although PARATY performs well on its own S-HTMP goals, there is no final proof of whether its performance on the original HTMP with uniform weights is superior to the dedicated Greedy+ baseline. Lack of a method to directly compare and minimize the standard average hit time under equal computational budgets.
3. Insufficient exploration of locality: Key business boundaries have not been fully discussed, including whether performance will degrade on non-strongly connected graphs.
4.  Unsubstantiated claims for directed graphs: The experiments for directed graphs only report PARITY's runtime. The critical claim of effectiveness lacks support, as no results on hitting-time reduction or comparisons to any baseline are provided for the directed setting.

**Questions:**

Please refer to the Weaknesses.

---

> ### Author Response · Authors · 2025-11-24
>
> We thank the reviewer for the constructive feedback. Below we reply to the points mentioned.
>
> 1. In the revised manuscript we have extended the discussion of the choice of $\alpha = \pi$, which was brief due to space constraints (see our paragraph after the definition of Problem 2).
> - First our choice of $\pi$ is motivated by the fact that real-world graphs are known to be fast-mixing.  That is, a random walk over $G_R$ will follow the transition probabilities of $\pi$ after a small number of steps.
> - Considering the fast-mixing behavior of random walks on real graphs, our weighting accounts for nodes that are more likely to be explored, i.e., nodes of higher relevance.
> - Computationally, our formulation can be solved in polynomial time, which we advocate as a significant advantage over other formulations.
>
>
> 2.
> - First, observe that we have already empirically evaluated the quality of the solution reported by PARITY on the uniform weighted HTMP ($f_{\text{avg}}$)(Section 5.1 - Q1).
> Our results show that the solution reported by PARITY on $f_{\text{avg}}$ is often comparable to or improves over the solution obtained by Greedy+.
> - Next, note that PARITY is an optimal algorithm for the S-HTMP objective with exactly $k$ edge additions.
> Instead, the Greedy+ baseline is a bi-criteria approximation algorithm for the HTMP objective with uniform weights selecting $O(k\ln(n/\varepsilon))$ edge additions to provide a $(2+\varepsilon)$-approximation. The main goal of our work is to optimize the S-HTMP objective: we do not aim at solving the original objective for the HTMP with uniform weights.
> However, in the revised manuscript, we provide a theoretical characterization of the solution of PARITY once evaluated for the HTMP with uniform weights. In the new Lemma 4, we show that there are classes of graphs such that the solution of PARITY is optimal for the HTMP with uniform weights.
> In addition, we empirically compare the solution reported by PARITY and the solutions reported by the baselines (see our common response).
>
> 3. As noted by the reviewer, we do not consider non-strongly connected graphs since such graphs do not admit a stationary distribution, and hence our problem formulation cannot be used. A practical solution for using our approach to non-strongly connected graphs is to execute PARITY on each strongly connected component of the graph.
>
> We kindly ask the reviewer to clarify the statement:
>
> > Key business boundaries have not been fully discussed.
>
> 4. We note that PARITY is the _first_ method able to optimize a variant of the HTMP objective on _directed_ graphs.  That is, the previous state-of-the-art of Adriaens et. al, can only optimize undirected graphs. We cannot therefore compare the solution of our algorithm PARITY with other baselines over directed graphs.

---

> > ### Comment · Reviewer_JVvJ · 2025-11-28
> >
> > My assessment aligns with other reviewers, focusing on the following key areas for improvement:
> > Performance boundary (or “performance envelope”): The constraints under which the system can achieve optimal performance. It is essential to fully discuss the boundary and the degree of descent outside of this condition through examples (as also noted by Reviewer mNfx). In addition, the research motivation should be clear and explicit. Finally, when the technical depth is limited, it is crucial to compensate by providing a comprehensive and in-depth experimental analysis.

---

> > > ### Author Response · Authors · 2025-12-03
> > >
> > > We thank the reviewer for the feedback and the clarification. We now answer the expressed concerns.
> > >
> > > > The constraints under which the system can achieve optimal performance. It is essential to fully discuss the boundary and the degree of descent outside of this condition through examples.
> > >
> > > - The solution provided by our new algorithm PARITY is optimal for the S-HTMP objective.
> > > - In our revised version of the work we prove a new result quantifying the perfomance of a solution $\mathbf{x}^\*$ obtained by PARITY once evaluated for the _average_ HTMP. Our new Lemma 4 quantifies the deviation between the objectives of the S-HTMP ($f_\pi(\mathbf{x}^\*)$) and the average HTMP ($f_{\mathrm{avg}}(\mathbf{x}^\*)$).
> > > - In addition, we empirically assess the tightness of our new result in Appendix D.6. As a summary:
> > >    - Theoretically there are classes of graphs for which PARITY obtains the optimal solution to the HTMP problem. For example, when $G_R$ corresponds to a $d$-regular graph;
> > >    - In practice the gap between $f_\pi(\mathbf{x}^\*)$ and $f_{\mathrm{avg}}(\mathbf{x}^\*)$ is often close to 0. However our theoretical upper-bound is often loose, leaving space for improved bounds.
> > >
> > > > In addition, the research motivation should be clear and explicit.
> > >
> > > - We rephrased Section 3 specifying explicitly and clearly the motivation of our new problem formulation (paragraph below Problem 2 in our revised version).
> > > - We provide applications for our problem formulation in the abstract, introduction and related work sections.
> > >
> > > > Finally, when the technical depth is limited, it is crucial to compensate by providing a comprehensive and in-depth experimental analysis.
> > >
> > > In our revised version, we supplement our already extensive experimental evaluation with additional scalability tests.
> > > In our new experiments, we test the performances of PARITY on graphs partitioned through the METIS algorithm. (see App. D.6. paragraph: Scalability on large data).
> > > In summary, PARITY can efficiently process very large graphs partitioned with METIS, confirming the results obtained through random partitions.

---

### Official Review · Reviewer_wbdc · 2025-10-31

**Soundness:** 3
**Presentation:** 2
**Contribution:** 1
**Rating:** 2
**Confidence:** 3

**Summary:**

This paper addresses the hitting time minimization problem, which consists of adding a fixed number of edges b to the graph to minimize the hitting time of a random walk process starting at a set of nodes R and ending at a set B. The paper considers a modification of the problem where the nodes in R are weighted by their value in the stationary distribution induced by R. They show that, different from the more general problem, the modified one can be solved in polynomial time using a greedy algorithm (PARITY). The experiments show (using 11 datasets) that PARITY achieves better values of the modified objective and is competitive with the best baselines for previously considered objectives (average and maximum hitting time). The experiments also show that PARITY is more efficient than the alternatives.

**Strengths:**

- The proposed problem formulation seems novel

- The proposed solution is simple and efficient

- The experiments consider multiple datasets and baselines

**Weaknesses:**

- The problem formulation is not well-motivated: It is not clear why the importance of nodes in R should be measured based on the stationary distribution induced by R (sometimes, the meaning of P_R is confusing, because it can also be the sub-matrix of P). I could see how someone might want to weight nodes in B because some of these nodes could be influential, for example. The paper also fails to provide how solutions to the new problem look compared to the original (uniform) problem using a few examples. In particular, I would be curious to see to what extent high degree nodes in R are favored in the solution of the new problem and how different the solution is from a simple degree heuristic. Some toy and/or small graph examples would be very helpful. Another point that was not clear to me is whether there is any restriction in the set of edges that will be added (sometimes that have to be edges connecting R to B, for example).


- The theory in the paper could be clearer for the reader: I did read the paper and checked the proofs but I feel like some of the arguments could be made much clearer. First, Lemma 2 basically claims that you can move the stationary weights inside the matrix inverse (so, I would assume that cannot be done with an arbitrary vector alpha). The proof of Lemma 2 does not state explicitly what is special about the stationary weights (in my opinion, this type of proof is much easier to follow in matrix form). Even the base case for Lemma 2 is not clear to me, because a matrix to the power zero is the identity, so the matrix power would lose information about X_0. Moreover, the proof of Lemma 3 doesn’t seem to use the fact that edges are selected based on Equation 5. More specifically, it not is not clear what property of Equation 5 is being leveraged beyond the fact that it is monotonically decreasing (this should not be sufficient). An example here could also be quite helpful.

- The fit of the paper to this conference is arguable: I don’t see how this paper falls into any of the topics of interest for ICLR. The closest topic I can find is learning on graphs. Looking at the citations, I see data mining, discrete optimization, social network venues but no paper from ICLR or similar conferences (e.g., ICML and NeurIPS).

**Questions:**

1) How is the problem formulation motivated by real-world applications?

2) How do solutions to the modified problem differ from those from the original problem in practice?

3) What is special about the stationary weights that makes Lemma 2 hold?

4) How does the proof of optimality in Lemma 3 depend on the specific form in Equation 5?

5) Why is this paper a good fit for ICLR?

---

> ### Author Response · Authors · 2025-11-24
>
> We thank the reviewer for the constructive feedback. Below is our response.
>
> 1.
> In our revised version of the manuscript, we better clarify the motivation for studying the S-HTMP objective. In particular,
> - for fast mixing graphs $G_R$, a random walk will follow the stationary distribution $\pi_R$ after a small number of steps. Fast mixing graphs are common in many domains, including social and web networks.
> - our formulation assigns more weights in its objective to nodes with higher probability in the stationary distribution over $G_R$.
> In real-world applications, this is motivated by the fact that a walk over $G_R$ will often start from those nodes with higher probability in the stationary distribution.
> For example, if we consider $G_R$ to be a Web graph, then search engines give a higher importance to web pages with higher PageRank value.
> In this example, let $R$ be the set of web pages with malicious content, and $B$ be the set of web pages with helpful content.
> First, note that when using a search engine, a user is likely to start from a node with a higher PageRank value.
> Next, our formulation suggests that to reduce the hitting time from pages in $R$:
>    - we should add links from nodes in $R$ that have a high PageRank value and are therefore explored more frequently; and
>    - among those nodes with high PageRank value we should favor nodes in $R$ poorly connected to $B$.
>
> 2.
> Our primary objective is to solve the S-HTMP problem, i.e., we do not solve S-HTMP as a means of solving a different problem. If the reviewers refers to how the solution to the S-HTMP provided by PARITY differs from the baseline solutions, we address this question in the revised version.
>
> In particular, in Appendix D.6 of the revised version, we provide a comparison between the solutions obtained by PARITY and the solutions obtained by baselines.
> We compare both the set of nodes $i\in R$ selected for new edge additions, and their average degree.
> As a summary, our algorithm PARITY reports significantly more different solutions compared to baselines:
> - the nodes $i\in R$ selected for edge additions by PARITY  are very different from those selected by our baselines (high Jaccard distance);
> - the nodes $i \in R$ in the solution of PARITY have high degree in $G_R$.
>
> Interestingly, as captured by our theoretical results, our algorithm does _not_ simply select the nodes with higher degree in $G_R$.
>
> 3.
> The key intuition behind Lemma 2 is explained below Lemma 2 (in lines 233-238 of the original manuscript). If $X_0 \sim \pi_R$, i.e., the Markov Chain follows its stationary distribution, then we do not need to take into account the topology of $G_R$, and we can obtain a simplified expression for the objective function.
>
> > Even the base case for Lemma 2 is not clear to me, because a matrix to the power zero is the identity, so the matrix power would lose information about $X_0$.
>
> The proof of the base case of Lemma 2 is trivial: note that $X_0 \sim \pi_R$, that is, $X_0$ is a node from $R$, yielding $Pr[\{X_0} \subseteq R] = 1$.
> Furthermore, note that the term $\mathbf{\pi_R^T D(x)A_R 1}$ is a scalar and _not_ a matrix.
>
> 4.
> Note that the specific form of Equation (5) is key for proving Lemma 3. That is, our algorithm PARITY selects link insertions of the form $(i, b)$ for $i$ maximizing $\Delta_i$ (lines 6 and 13 of the pseudocode of PARITY) at each iteration. Note that $\Delta_i$ corresponds to the reduction of the contribution of $i\in R$ in Equation (5). We use the fact that at each iteration PARITY will select a node with the highest $\Delta_i$ (which depends on Equation (5)) to prove the optimality of our algorithm in Lemma 3.
>
> 5.
> We address this issue in our common response and in the revised manuscript.
>
> Finally, we comment on the following points mentioned by the reviewer.
>
> > Another point that was not clear to me is whether there is any restriction in the set of edges that will be added (sometimes that have to be edges connecting R to B, for example).
>
> The objectives for HTMP and S-HTMP are defined over link-insertion vectors.  A link-insertion vector (see lines 133-140 in the original manuscript) is defined over (non-existing) edges of the form $(i, b)$ for $i\in R$ and $b \in B$. That is, the link insertions that we consider have the form $(i, b)$ for $i\in R$ and $b\in B$. We have highlighted this in the revised manuscript.
>
> > I could see how someone might want to weight nodes in B because some of these nodes could be influential, for example.
>
> Note that our Observation 1 highlights the fact that the HTMP objective does _not_ depend on the specific node $b\in B$ of the new edge addition $(i, b)$, but only on the node $i\in R$. We agree with the reviewer that accounting for the importance of nodes in $B$ is an interesting research question; requiring a different and new problem statement. Note that PARITY can be used by reversing the roles of $R$ and $B$, when $B$ corresponds to the group of interest.

---

### Author Response · Authors · 2025-11-24
**Common response**

We thank all reviewers for their insightful feedback that helped us improve the quality of our manuscript.
We have uploaded a revised version together with our rebuttal.
In our revised version, we mark all edits with respect to the original version submitted in blue.

We now first address a major concern that emerged regarding the fit of our work to the ICLR conference.
Then, we will summarize the edits in our revised version of the work.

**Relevance to ICLR** (reviewers wbdc and fDD5)

We acknowledge that the topic of our paper is not listed _explicitly_ in the call for papers. However, the call mentions that the list is _non-exhaustive_.
The study of foundational combinatorial problems related to graph problems and graph metrics is highly relevant for machine learning and deep learning.
Evidently, several works on combinatorial optimization for graph problems are published each year in ICLR, ICML, and NeurIPS conferences.

By quoting from the call for papers:
> ICLR is globally renowned for presenting and publishing cutting-edge research on all aspects of deep learning used in the fields of artificial intelligence, statistics and data science, as well as important application areas such as machine vision, computational biology, speech recognition, text understanding, gaming, and robotics.

To address this concern, we have extended the related work section of our manuscript as follows:
> For machine-learning applications, the optimization of several hitting-time metrics through $k$ edge additions has been studied in the context of Markov Decision Processes [1+,2+], and GNN over-squashing and over-smoothing prevention [3+,4+].

[1+] Discovering options for exploration by minimizing cover time, Jinnai et al., ICML, 2019.
[2+] Exploration in reinforcement learning with deep covering options, Jinnai et al., ICLR, 2020.
[3+] DiffWire: Inductive Graph Rewiring via the Lovasz Bound, Arnaiz-Rodriguez et al, PMLR, 2022.
[4+] Understanding oversquashing in GNNs through the lens of effective resistance, Black et al., ICML, 2023.

Note that our problem formulation is in fact _general enough_, and can be of interest to all of the above settings.
Especially given the guarantees offered by our algorithm and its extremely high efficiency.

**Major edits to the manuscript**

In the following list, we use W\*.Y to denote the weakness/question number \* pointed out by reviewer Y.

1. We reinforce the motivation for studying the S-HTMP problem in Section 3 (W1.wbdc, and W1.JVvJ).
2. We provide a theoretical connection between the S-HTMP objective and the uniform HTMP objective, for the solution obtained by our algorithm PARITY; formalized in Lemma 4 (W3.fDD5 and W2.JVvJ).
3. We empirically compare the properties of the solutions returned by PARITY and baselines GREEDY+ and RepBubLik.
We evaluate two different metrics, the Jaccard distance and the average degree of the nodes in $i\in R$ selected in the respective solutions (Section 5.1 and Appendix D.6) (W1.wbdc).

Finally, we addressed all other minor concerns and typos noted in individual reviews.

Obviously, we are willing to further improve the quality of our manuscript based on the further constructive feedback by the reviewers.

---

### Author Response · Authors · 2025-12-03
**Summary**

We thank all reviewers for their feedback that improved substantially the quality of our manuscript. We now briefly summarize how we addressed _all_ concerns over the rebuttal period.

- We use W\*.Y to denote weakness/question \* by reviewer Y in their _Official Review_;
- We use (W\*).Y.C to denote weakness/question \* by reviewer Y in _Official Comment_ number C (chronological ordering assumed); where (W\*) denotes an _optional_ field.

1. Relevance to ICLR (w5.bdbc, w1.fDD5, w1.fDD5.1).
    - we revised the introduction, related work and conclusion sections. We now provide applications for machine learning tasks of our work, such as graph neural network design.
2. Theoretical guarantees of the optimal solution of our new objective (S-HTMP) and the objectives studied in literature (average, max HTMP). (w2.JVvJ,JVvJ.1,w3.fDD5,w3.fDD5.1).
    - we provide a new lemma (Lemma 4) upper-bounding the distance between the solution obtained by our algorithm PARITY once evaluated on the S-HTMP and the average HTMP. Our bound depends on the maximum hitting time.
    - we empirically validate our new Lemma 4 (app. D.6). (w3.fDD5.1).
3. Motivation for studying our new objective (S-HTMP) and technical depth. (w1.wbdc, w1.JVvJ, JVvJ.1, w2.fDD5, w3.fDD5, w4.fDD5.1, w3.mNfx).
   - we strengthened the motivation of our new objective in the introduction and Section 3.
   - _theoretically_ our new result in point 2. above highlights the following: our problem objective shows that there are classes of graphs for which our algorithm PARITY provides _optimal solutions to the average HTMP_.
   - _in practice_ our algorithm requires almost linear-time scaling on massive dataset and can process directed graphs. Current state-of-the-art methods cannot match our guarantees given that they optimize more complex metrics, and do not scale (as shown in our experiments).
4. Additional empirical validation.
   -  we empirically compare the solutions of our algorithm compared to the solutions of previous methods. (App. D.6) to charaterize the difference between PARITY's solution and the solution of existing methods (w2.wbdc).
   -  we provide experiments using partitions obtained with the popular METIS algorithm to show the scalability on non-random partitions (w2.mNfx, w2.mNfx.1, w3.JVvJ, JVvJ.1).
5. Writing and presentation.
   - we improved the clarity of the problem statement
   - we fixed the notation, existing typos, and adopted standard terminology (w4.fDD5, w2.fDD5.1, w3.fDD5.1)

Finally, we clarified in our Official Comment various doubts posed by individual reviewers requiring no edits of our manuscript. (w3.wbdc, w4.wbdc, w4.JVvJ, w1.mNfx, w3.mNfx)

---

### Meta-Review · Area_Chair_zAa9 · 2026-01-12

**Summary:**

There are multiple concerns which I am not going to repeat here, just for the sake of repeating them.
The reviewers have clearly stated their concerns.

The most important is that the paper is not well motivated currently. The authors need to put some effort in re-writing the paper.

**Reviewer Concerns:**

To be fair, the authors have put significant effort in addressing the concerns. However, the paper needs a major revision.

**Reviewer Scores:**

All reviewers replied except of one, and they are negative.

---

### Decision · Program_Chairs · 2026-01-26

Reject